



# Influences of lidar scanning parameters on wind turbine wake retrievals in complex terrain

Rachel Robey[1] and Julie K. Lundquist[2,3]

[1]Department of Applied Mathematics, University of Colorado Boulder, Boulder, Colorado, USA
[2]Department of Atmospheric and Oceanic Sciences, University of Colorado Boulder, Boulder, Colorado, USA
[3]National Renewable Energy Laboratory, Golden, Colorado, USA

**Correspondence:** Rachel Robey (rachel.robey@colorado.edu)

**Abstract.** Scanning lidars enable the collection of spatially resolved measurements of turbine wakes and the estimation of wake properties such as magnitude, extent, and trajectory. Lidar-based characterizations, however, may be subject to distortions due to the observational system. Distortions can arise from the resolution of the measurement points across the wake, the projection of the winds onto the beam, averaging along the beam probe volume, and intervening evolution of the flow over the scan duration. Using a large-eddy simulation and simulated measurements with a virtual lidar model, we assess how scanning lidar systems may influence the properties of the retrieved wake using a case study from the Perdigão campaign. We consider three lidars performing range-height indicator sweeps in complex terrain, based on the deployments of lidars from the Danish Technical University (DTU) and German Aerospace Center (DLR) at the Perdigão site. The unwaked flow, measured by the DTU lidar, is well-captured by the lidar, even without combining data into a multi-lidar retrieval. The two DLR lidars measure a waked transect from different downwind vantage points. In the region of the wake, the observation system interacts with the smaller spatial and temporal variations of the winds, allowing more significant observation distortions to arise. While the measurements largely capture the wake structure and trajectory over its 4-5 $D$ extent, limited spatial resolution of measurement points and volume averaging lead to a quicker loss of the two-lobes in the near wake, smearing of the vertical bounds of the wake (<30 m), wake center displacements up to 10 m, and dampening of the maximum velocity deficit by up to a third. The virtual lidar tool, coupled with simulations, provides a means for assessing measurements capabilities in advance of measurement campaigns.

## 1 Introduction

Scanning lidars are increasingly employed in the collection of spatially resolved measurements of wind turbine wakes, which are characterized by reduced wind speeds and increased turbulence. Lidar measurements may be used to diagnose key metrics such as the magnitude of the velocity deficit, the spatial extent of the wake, and the trajectory followed by the wake. Building from promising early deployments for wake measurements in the early 2000s (Käsler et al., 2010), scanning lidars have become a staple in many field campaigns for assessing wind turbine wakes (Aitken et al., 2014b; Aitken and Lundquist, 2014; Barthelmie and Pryor, 2019; Bodini et al., 2017; Brugger et al., 2020; Gottschall, 2020; Iungo et al., 2013; Menke et al., 2018b;





Smalikho et al., 2013; Wildmann et al., 2018b) and other complex flow features like recirculation zones (Menke et al., 2019a)
or urban flow features (Newsom et al., 2008).

Investigations of turbine wakes with lidar were initially conducted in fairly simple terrain (e.g. Käsler et al. (2010); Aitken et al. (2014b); Bodini et al. (2017)). For a detailed summary of lidar measurements of wakes, see Gottschall (2020). The Perdigão field campaign was a seminal study of the behavior and measurement of turbine wakes in complex terrain (Fernando et al., 2019); the site features parallel double ridges with a single 2 MW turbine, the wake of which was captured through
an extensive array of scanning lidar (Fig. 1). The wake behavior in the highly complex site using the lidar data and different scanning strategies are explored in Wildmann et al. (2018b), Menke et al. (2018a), and Barthelmie and Pryor (2019).

Wake characterizations based on lidar data, however, may be subject to distortions due to inherent properties of the observational system. Even assuming perfect calibration and sufficient air quality to ensure robust returns, the measurements are subject to the spatial resolution of retrieval points, projection of the winds onto the lidar beam, averaging along the beam
probe volume, and intervening evolution of the flow over the scan duration. Possible measurement and scan techniques have grown in complexity in part to address some of these issues, e.g., combining multi-lidar measurements to correct for projection (Wildmann et al., 2018b; Vasiljević et al., 2017) and automated synchronization to follow the wake meandering (Barthelmie and Pryor, 2019). Additionally, progress in deployments of scanning lidar and work with observations has begun to be supported by "virtual lidar" models acting on simulations of waked flow. Doubrawa et al. (2016) investigate how the spatially and
temporally disjunct nature of the lidar wake measurements affect the retrieval and highlight the need for spatial distribution of the sampled points. Meyer Forsting et al. (2017) focus on the effect of averaging over the lidar probe volume, noting the strongest influence around the wake edges.

In this study, we continue in this vein using a large-eddy simulation (LES) and virtual instrument model (Robey and Lundquist, 2022) to assess how the configuration of scanning lidar systems influences the properties of the retrieved wake
in a realistic case study from the Perdigão campaign. We consider stand-alone range–height indicator (RHI) scans of transects across the Perdigão ridges and leverage the simulated measurements to perform novel analysis disaggregating and attributing the causes of potential distortions in the measured wake. Section 2 describes the Perdigão site, selected case studies and data processing, as well as the setup of the LES and virtual lidar models. Section 4 describes how the collected wake data are processed into vertical profiles and fitted to quantify the wake trajectory, strength, and extent. Section 5 presents results of the
virtual LES measurements and distortions and goes on to compare measurements from real observations. A discussion of the results and summary of the key conclusions are given in Sects. 6 and 7.

## 2 Perdigão campaign and selection of case studies

The Perdigão field campaign, conducted near the Vale do Cobrão in central Portugal, comprised an extensively instrumented investigation of flow over parallel ridges with the intensive observation period spanning from May to June 2017 (Fernando
et al., 2019). Dominant climatology at the site features winds perpendicular to the 2 km long ridges. Combined with the symmetry of the ridges and valley, the winds resemble that of idealized, two-dimensional valley flow. Scanning lidars deployed



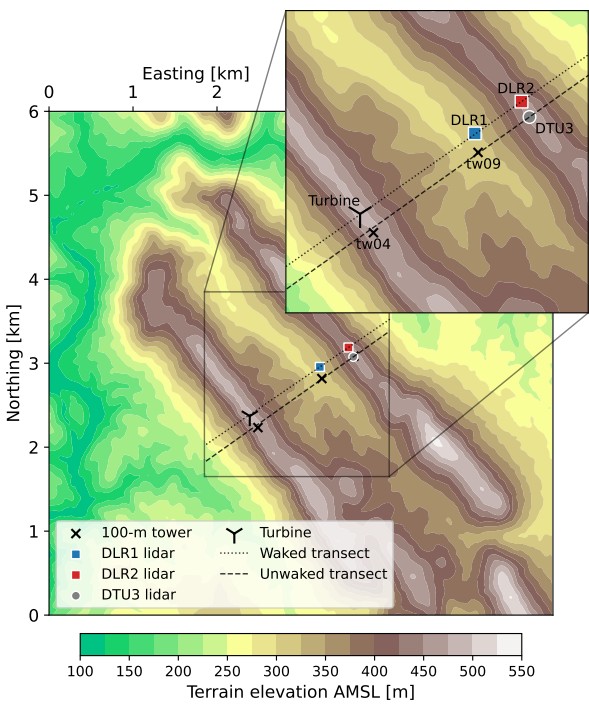

**Figure 1.** Plan-view topographic map of the Perdigão site over the inner LES domain. Inset of the valley details locations of the turbine, 100 m towers, and the scanning lidars with corresponding transect planes.

**Table 1.** Summary of selected case study periods

|  | Case 1 | Case 2 | Case 3 |
|---|---|---|---|
| Start time [UTC] | 21-05 21:30 | 13-06 4:30 | 14-06 0:00 |
| End time [UTC] | 05-22 1:30 | 06-13 7:00 | 06-14 7:30 |
| Duration [h] | 4 | 2.5 | 7.5 |
| Wind speed [m s$^{-1}$] | 6.1 | 5.7 | 5.6 |
| Wind direction [deg] | 218 | 218 | 218 |

at the site provide insight into the spatial structure of flow dynamics and phenomena. The collected data reflect atmospheric processes appropriate to the regime, including low-level jets, stable mountain waves, and recirculations in the valley (Menke et al., 2019a). On the upwind ridge is a single 2 MW Enercon E-82 turbine (78 m hub height, rotor diameter $D = 82$ m), the

wake of which was captured as it propagated over the valley in southwesterly flow conditions.

We refine a set of stable case studies with winds at operating speeds perpendicular to the ridges so that a significant portion of the wake may be expected in the scanning transect. Using 30-minute averaged data from the upwind ridge tower (tw04, Fig. 1), the data are filtered such that the 10 m winds are perpendicular to the ridges (southwesterly, 200°-230°) and at turbine



WIND
ENERGY
SCIENCE
DISCUSSIONS

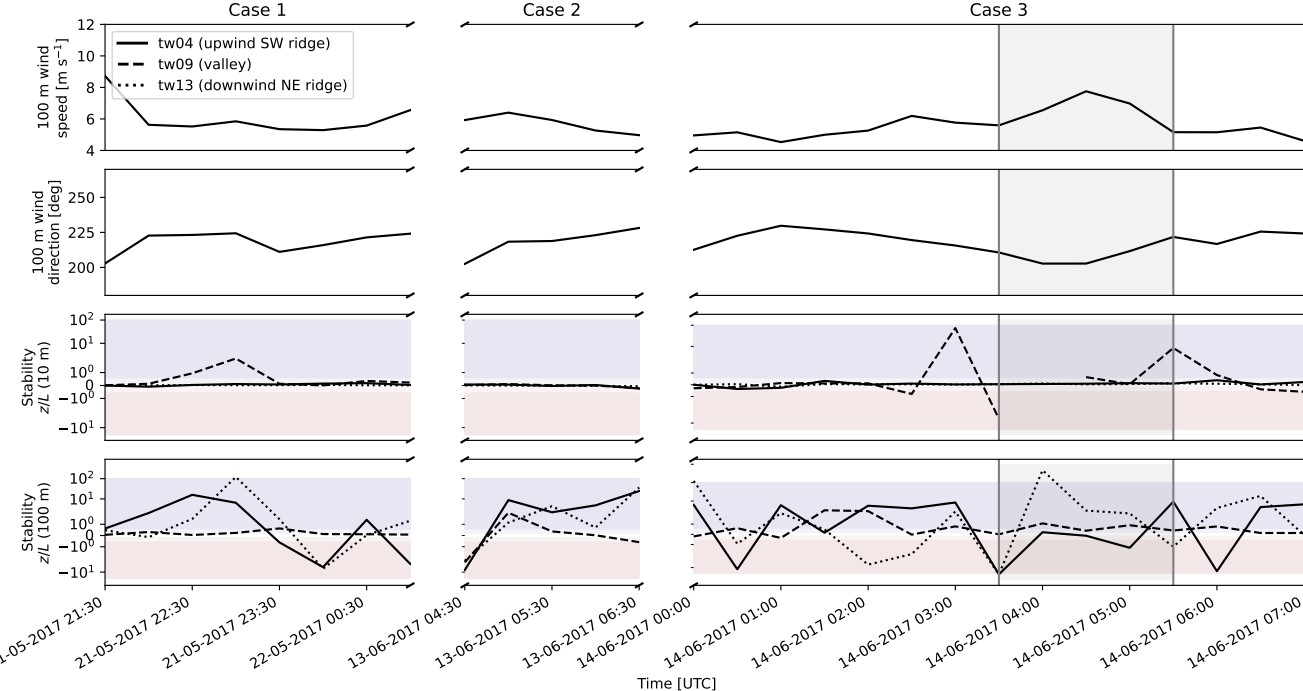

**Figure 2.** Thirty-minute averaged wind and stability metrics for the three selected case studies. Red and blue shading indicate areas in which the Obukhov stability parameter is unstable and stable, respectively. The gray shaded region within Case 3 indicates the simulation period.

operating speeds (>4.5 m s$^{-1}$, cut in at 2 m s$^{-1}$). We identified three case study periods with good lidar data that match these

conditions for more than two consecutive hours (Table 1, Figure 2). The final case overlaps with the high-resolution Weather Research and Forecasting (WRF) LES period from 3:30 to 5:30 UTC on 14 June 2017 (Sect. 3.1).

Stability conditions are diagnosed from the Obhukov stability parameter, $z/L$, at three towers across the ridges and valley at 10 m and 100 m (Fig. 2). The Obukhov length, $L$, is defined as

$$L = -\frac{u_*^3 \Theta_0}{\kappa g \overline{w'\theta'}} \tag{1}$$

where $u_*$ is the friction velocity, $\Theta_0$ is the surface potential temperature, $\kappa = 0.4$ is von Kármán's constant, $g = 9.81$ is the acceleration due to gravity, and $\overline{w'\theta'}$ is the heat flux. The dimensionless $z/L$ parameter, which has been computed over half-hour periods, indicates stability following the classifications in Rodrigo et al. (2015), symmetrically extending those of Sorbjan and Grachev (2010): nearly neutral (|z/L| < 0.02), weakly stable ( 0.02 < z/L < 0.6), and very to extremely stable (0.6 < z/L). Over the complex topography of the site, significant variations in the stability parameter can occur, reflected by the variation in

the values reported at the different towers and heights. The selected cases highlighted here are mostly neutral to stable (Fig. 2).

From the available instrumentation, we narrow our focus to three Leosphere WindCube 200S systems performing vertical-slice RHI scans of transects across the ridges (Fig. 1). The positions and scan configurations of the three Danish Technical





**Table 2.** Placement and scan parameters of the three lidar systems

|  | DTU LRWS3 | DLR1 | DLR2 |
|---|---|---|---|
| Instrument height AMSL [m] | 452.3 | 323.2 | 458.2 |
| Instrument position | 39° 42' 48.69" N | 39° 42' 44.83" N | 39° 42' 52.31" N |
|  | 7° 43' 49.68" W | 7° 44' 6.41" W | 7° 43' 52.12" W |
| Instrument placement | unwaked | waked | waked |
|  | downwind ridge | valley | downwind ridge |
| Azimuth angle $\theta$ [deg] | 235 | 237 | 237 |
| Minimum elevation angle $\phi_0$ [deg] | -6 | 6 | -12 |
| Maximum elevation angle $\phi_N$ [deg] | 30 | 160 | 90 |
| Elevation angle interval $\Delta\phi$ [deg] | 0.75 | 1 | 1 |
| Beam accumulation time [s] $\Delta\phi$ [deg] | 0.5 | 0.5 | 0.5 |
| Scan sweep duration [s] | 24 | 77 | 51 |

University (DTU) and German Aerospace Center (DLR) lidars analyzed in this study are detailed in Table 3 (Mann, 2019; Wildmann, 2019). Two lidars operated by DLR scan a transect intersecting the turbine (Wildmann et al., 2018a). With one

instrument in the valley and the other on the northwest ridge (Fig. 3), the DLR scans collect data on the waked flow in the perpendicular transect from different vantage points. Note that in the archive the DLR lidar on the downwind ridge was swapped on 3 June 19:20 UTC from instrument 85 to 89; the lidar at this position will be labeled as DLR2 throughout, even though the instruments were switched. The final WindCube 200S is part of the DTU long-range wind scanner system (Vasiljević et al., 2016; Menke et al., 2019b) and covers a comparable, unwaked transect separated from the turbine transect by ~125 m.

Alternate scanning approaches are possible, including horizontal cuts through the wake and other coordinated scans deployed at Perdigão, that are not treated here. In fixed transect scans such as the ones we use, meandering of the wake can impede measurements as the bulk of the wake may move out of the observed transect. By constraining the cases to those perpendicular to the ridges, we increase the expectation that a substantial portion of the wake is present in the transect. Acknowledging that the reference truth of the winds in the transect may not represent the full wake in the three-dimensional space, our analysis

focuses on the potential observational distortion of the wake as it exists in the transect.

To control the quality of the lidar data, measurements from all instruments are filtered by the carrier-to-noise-ratio (CNR), requiring CNR < -24 dB. The scans are further averaged over 5 minutes before use (see Sect. 4.1).



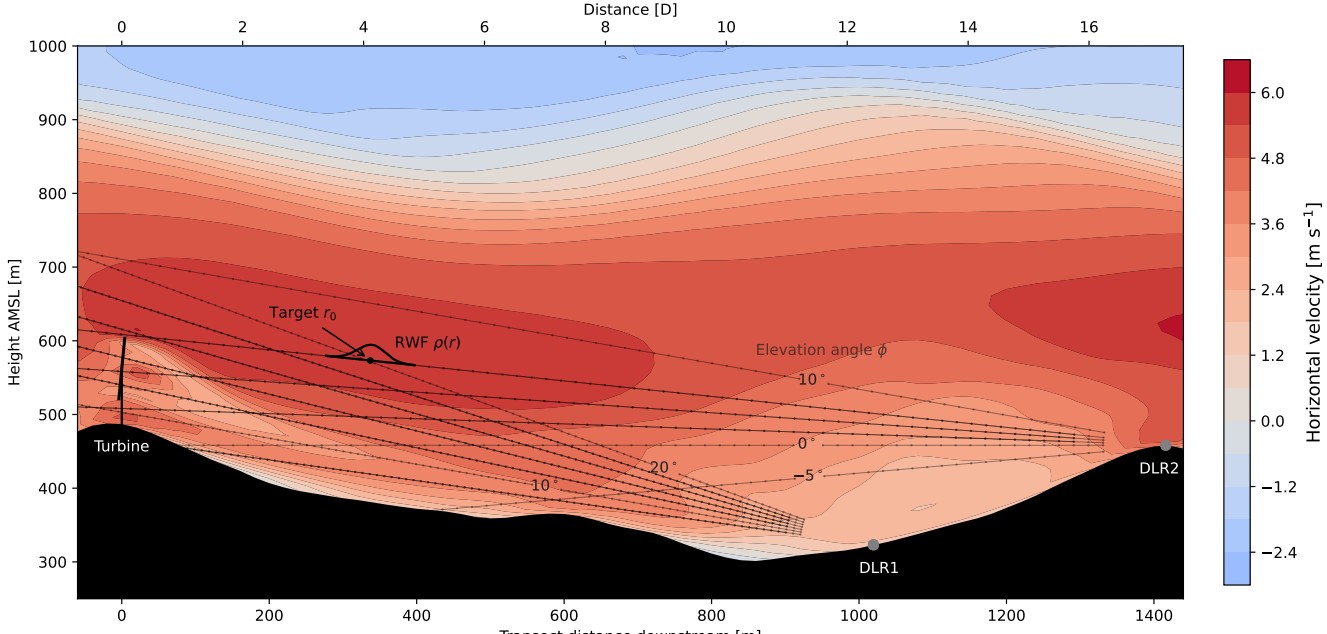

**Figure 3.** Waked transect cross section with LES horizontal winds and the layout of the turbine and DLR lidars. Sample lidar beams with elevation angles, target retrieval points, and range-gate weighting function (RWF) probe scale are shown for reference. Top axis shows distance in rotor diameters, $D$.

# 3 Simulation of flow and lidar instrumentation

## 3.1 Case study simulation

To better understand the dynamics and interaction of the lidar systems with the flow, we employ a simulation of the flow field during 2 hours of the 14 June 2017 case study combined with a model of the instrument retrieval. The simulation uses the WRF LES model (Skamarock et al., 2019) and is a reproduction of a validated simulation of the case study with the model (Wise et al., 2022). We follow the configurations therein, updated for WRFv4.3 mesoscale-to-microscale (MMC) model and outputting high-frequency (1 Hz) output on which to run the virtual lidar model. As in Wise et al. (2022), five domains are used to nest down from mesoscale, forced by Global Forecast System reanalysis data (National Centers for Environmental Prediction, National Weather Service, NOAA, U.S. Department of Commerce, 2015), down to fine LES scales. The nests refine the horizontal resolution from the outer mesoscale domains with 6750 m and 2250 m horizontal grid size to increasingly fine LES resolution with 150 m, 50 m, and finally 10 m grid spacing in the innermost domain. The outer two domains were allowed to spin up for 9 hours before starting the inner domains. These in turn spin up for an additional 30 minutes before the simulation data are used.



**Table 3.** WindCube 200S lidar parameters

| | |
|---|---|
| Fast Fourier Transform Points $M$ [#] | 64 |
| Digitization frequency $f_s$ [MHz] | 250 |
| Range gate $\tau_m$ [ns] | 256 |
| Pulse full width at half maximum $\tau$ [ns] | 200 |
| Probe full width at half maximum $\Delta p$ [m] | 44 |

For land input, we use 1 arcsec terrain from the Shuttle Radar Topography Mission (Farr et al., 2007) and 100 m land use data from the CORINE Land Cover 2006 raster dataset (Bossard et al., 2000). We follow adjustments in Wise et al. (2022) to the roughness length of mixed shrubland/grassland to 0.5 m. The turbine feedback on the flow in the finest domain is represented via the generalized actuator disk model in the MMC release (Mirocha et al., 2014). The lift and drag coefficients for the 2 MW

E-82 Enercon turbine at the site are not publicly available, so a representative turbine with similar parameters is used in the simulation (Arthur et al., 2020).

Inner-domain LES winds are output every second, and the winds on the waked and unwaked transect planes are extracted as a "truth" reference against which to compare the virtual lidar measurements. To represent the raw velocity fields in as unadulterated a way as possible, the velocities are interpolated in the horizontal directions to positions along the transect, with

the vertical grid unchanged (Fig. 3). The stretched vertical grid provides sufficient resolution across the wake with 20 points in the lowest 200 m above the surface.

## 3.2    Virtual instrument model and configuration

To represent the lidars in the simulation, virtual instruments are placed in the finest LES domain following deployment locations and scan parameters from the campaign (Table 3). The virtual lidar model mimics the wind retrieval and scanning pattern of

the instrument; details can be found in Robey and Lundquist (2022), but we briefly summarize the key points here.

Coherent Doppler lidars measure wind speeds at target distances by diagnosing the Doppler shift in laser light that is backscattered off of suspended aerosols. Scanning lidars perform an RHI scan by sweeping their beam through a vertical slice of varying elevation angle in time. For the simplified virtual model, we assume uniform and adequate aerosol distribution, omitting aerosol type, size, and density distribution and the influence of conditions like humidity, fog, or precipitation on the

return signal (Aitken et al., 2012; Boquet et al., 2016; Rösner et al., 2020).

The virtual lidar model can be considered as a series of stages: interpolation/selection at the location of the retrieval points, projection of the velocities onto the elevated beam, averaging over the the probe volume, and advancement of the beam position over the scan duration. References for the beams, retrieval points, elevation angles, and the probe volume weighting are shown in Fig. 3. Note that each of the stages performs a linear operation on the wind field, and error in the estimated horizontal

velocity incurred by each is directly additive. The impact of each of the stages on the retrieval may be separated out via partial models using only a subset of the stages.





*(1) Interpolation.* Once the lidar geometry (position and scanning angles) sets the beam location, the wind components are interpolated to points along the lidar beam using linear barycentric interpolation from a Delaunay triangulation of the LES grid. At the retrieval points, this is what the lidar would measure if it could perfectly collect 3D winds.

*(2) Projection.* The wind velocity vector, $\boldsymbol{u} = (u, v, w)$, is projected onto the beam unit direction vector, $\hat{\boldsymbol{b}}$. The lidar senses only the radial (line-of-sight) velocity, $v_r$ (Eq. 2).

$$v_r = \hat{\boldsymbol{b}} \cdot \boldsymbol{u} = u_h \cos\phi + w \sin\phi \qquad (2)$$

where $u_h = u \sin\gamma + v \cos\gamma$ is the horizontal velocity in the transect of the azimuthal angle, $\gamma$, and $\phi$ is the elevation angle of the beam above the horizon. Under this convention, positive radial velocities move away from the instrument.

*(3) Range-gate weighting (RWF).* Due to the lidar measurement process, the radial velocity measured by the lidar at target distance $r_0$, $\overline{v}_r(r_0)$, is not a point value but an average of winds in a probe volume along the beam. The averaging is well-represented by a convolution of the projected wind velocities along the beam with a range-gate weighting function, $\rho(s)$ (Eq. 3).

$$\overline{v}_r(r_0) = \int_{-\infty}^{\infty} \rho(s) v_r(r_0 + s) ds \qquad (3)$$

Both DTU and DLR systems considered in this study are WindCube 200S lidar; we correspondingly use a model for a pulsed RWF (Eq. 4), based on the convolution of the pulse with the range-gate observation window (Banakh and Smalikho, 1997; Cariou and Boquet, 2010).

$$\rho(r) = \frac{\text{erf}\left(\frac{4\sqrt{\ln 2}}{c\tau}r + \frac{\tau_m \sqrt{\ln 2}}{\tau}\right) - \text{erf}\left(\frac{4\sqrt{\ln 2}}{c\tau}r - \frac{\tau_m \sqrt{\ln 2}}{\tau}\right)}{c\tau_m} \qquad (4)$$

Here, $c$ (0.29979 m ns$^{-1}$) is the speed of light and $\tau = 200$ ns is the full-width half-maximum (FWHM) of the Gaussian pulse

used by lidar system. The range-gate observation time, $\tau_m$, arises from the window for the fast Fourier transform (FFT) used for the frequency diagnosis. With $M$ points of a signal digitized at a frequency of $f_s$, the range-gate window is $\tau_m = M f_s^{-1}$. We use consistent sampling settings across all three WindCube 200S based on the values reported in the DTU lidar data files and summarized in Table 3 (Mann, 2019). The temporal window corresponds to a spatial probe length, $\Delta p$, over which the contributions to the signal originated. An estimate of $\Delta p$ is given in Banakh and Smalikho (1997) as the estimate of the FWHM

of the RWF (Eq. 5).

$$\Delta p \approx \frac{c\tau_m}{2\text{erf}(\sqrt{\ln(2)}\frac{\tau_m}{\tau})} \qquad (5)$$

For this system, $\Delta p \approx 44$ m, which aligns with visual inspection of the modeled RWF curve (shown for reference in Fig. 3). Probe lengths of 30 m have been reported by Menke et al. (2019a), which may arise from altered parameters or different approaches to the RWF model and estimation.



In the virtual lidar model, the weighting integral of the radial velocities by the RWF (Eq. 3) is approximated by a discrete weighted average using interpolated velocities at positions $s_k$ along the beam, spaced every 1 m (Eq. 6).

$$\overline{v}_r(r_0) \approx \sum_k \frac{h_k \rho(s_k - r_0)}{\sum_i h_i \rho(s_i - r_0)} v_r(s_k) \tag{6}$$

When the probe volume intersects with a hard object, the strike can corrupt the wind data; the model accounts for blocking of the beam due to terrain by requiring valid points reaching 80 % of the RWF volume. Contamination due to hard strikes of the turbine, which do occur in the actual data, are not replicated.

*(4) Time-staggering.* The sweep of the beam over the duration of the scan is realized in time by staggering the retrieval of the radial velocities over the high-frequency (1 Hz) LES output. Some of the real instruments adjusted the beam every 0.5 s, and linear interpolations between the LES output are used for these intermediate times. We further note that this model does not currently include continuous scanning collecting data from over the arc of travel; the beam retrievals reflect fixed, incremented positions.

## 4 Wake fitting

### 4.1 Construction of deficit vertical profiles

Our analysis concerns distortions of the wake in the two-dimensional transect plane. In order to quantify and characterize the wake and allow for comparison of true and observed behavior, we construct vertical profiles of the horizontal velocity deficit downwind of the turbine. Construction of the vertical deficit profiles entails estimation of the horizontal velocities from line-of-sight velocities, interpolation of the irregular data to the vertical profiles, and time averaging to allow for a robust comparison between waked and unwaked transects to isolate a wake deficit. Processing of the wake is identical for virtual and (quality-controlled) observational data.

To estimate the horizontal velocity from radial velocities, $v_r$, the projection of the vertical velocity is assumed to be negligible, meaning that the radial velocities are purely a projection of the horizontal velocity. Equation 7 solves for the horizontal velocity under this assumption.

$$u_{h,\text{lidar}} = \frac{v_r}{\cos \phi} = u_h + w \tan \phi \tag{7}$$

where $\phi$ is the beam elevation angle. The assumption is robust so long as $u_h \gg w \tan \phi$. Significant vertical velocities do occur given the complex terrain and wake, but the elevation angles of the beams scanning the upwind ridge in the area of the wake are kept small, $\phi < 20°$ ($\tan 20° \approx 0.36$), constraining the potential contamination from vertical velocities when relying on a single lidar.

We prescribe vertical profiles along the transect every 20 m ($\sim D/4$) downstream of the turbine with 2 m resolution vertically. The lidar-recovered horizontal velocities are linearly interpolated to the profiles from the irregular retrieval points using a Delaunay triangulation as in Iungo and Porté-Agel (2013). Vertical profiles for the unwaked transect are also computed.





To determine the wind deficit, we leverage the symmetry of the campaign site and define the unwaked transect to be the freestream velocity, $u_{h,\text{unwaked}}$. To obtain a more robust profile and temper localized variations in the flow between transects, the data are time-averaged. We found, with both LES and observational data, that a 5-minute average is sufficient to find good agreement between the vertical profiles in the waked and unwaked transects and produce well-defined wake deficits. The period corresponds to about 3 RHI sweeps for DLR1, 6 for DLR2, and 13 for the reference DTU3 (and 300 snapshots in the 1 Hz LES wind field). Only full RHI lidar sweeps are used, and any sweep with time stamps inside the window are included. Longer averaging windows (e.g., 10 or 30 minutes) are possible, but a shorter window preserves more of the fast dynamics of the wake.

Aligning the transects, we compute the velocity deficit (Eq. 8) from the corresponding time-averaged vertical profiles of $u_h$:

$$\Delta u_h = \left( 1 - \frac{u_{h,\text{waked}}}{u_{h,\text{unwaked}}} \right) \times 100 \ \% \tag{8}$$

where $\Delta u_h$ is the percent reduction in wind speed.

The true LES wake deficit is also defined by the difference in flow between the waked and unwaked transects so that wake errors arise purely from the influence of the observation system. Here, the raw representation of LES flow field is designed to be as minimally processed as possible; the winds are only horizontally interpolated to the locations of the vertical profiles along the transect and left on the native vertical grid. Time averages occur across constituent 1 Hz output and determination of the deficit is performed via a difference between waked and unwaked transects as with the lidar data. The profile construction process standardizes the LES, virtual measurements, and observations into the same format with the deficit profiles reflecting the wake.

## 4.2 Wake-fitting algorithm

To distill and quantify the behavior of the wake and facilitate inter-comparison, both against the LES truth and between measured wakes, we fit Gaussian models (Eqs. 9,10) to the vertical profiles and extract the magnitude, center height, and vertical extent of the wake following the approach in Aitken et al. (2014a).

As the winds flow past the turbine and interact with the aerofoil blades, we can sketch the wake. Maximum drag on the wind field typically occurs near the blade tip, leading to two distinct local maxima in the wind deficits close to the rotor (the near wake) (Martínez-Tossas et al., 2015). As these lobes propagate downstream, they expand and mix with the ambient turbulent flow and eventually merge and become indistinguishable (the far wake). The deficit diminishes and the wake finally dissipates. To capture the range of behavior over the wake extent, a combination of Gaussian models are used.

In the far wake where the lobes have merged, the vertical deficit profile is modeled as a single Gaussian (Eq. 9).

$$g_1(z) = a \exp \left[ -\frac{1}{2} \frac{(z - z_c)^2}{w^2} \right], \tag{9}$$

Here, $a$ is the maximum magnitude of the deficit and $z_c$ is the vertical location of the center of the wake. The width of the Gaussian is controlled by the parameter $w$; we take the corresponding vertical extent (width) of the wake to be the full width at which $g_1$ has decayed to 5 % of the maximum, i.e., $2\sqrt{2\ln(20)}w$.

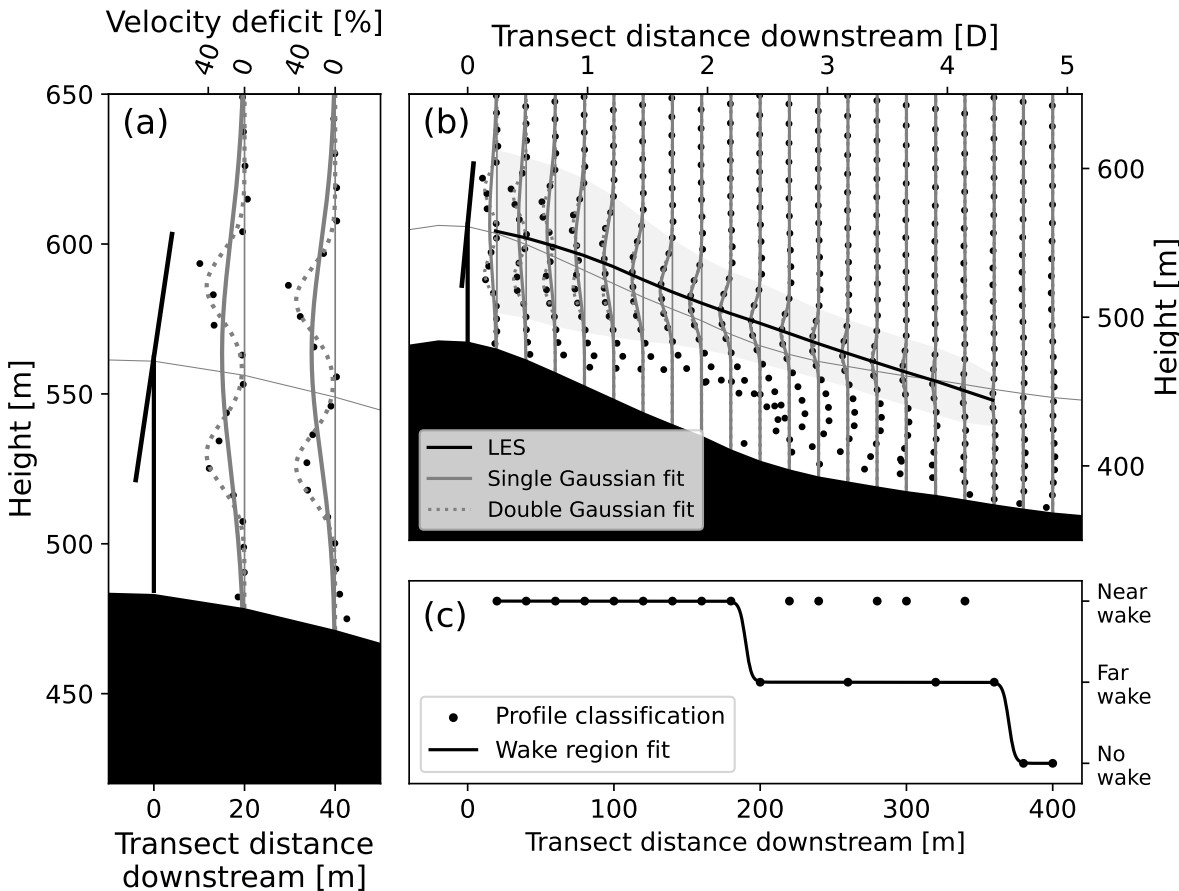

**Figure 4.** Example wake fit. Single- and double-Gaussian wake models fitted to (a) first few vertical wind deficit profiles and (b) over the larger wake, moving left to right, and highlighting the wake center and extent. (c) Classifications of the individual vertical profiles are fitted into cohesive wake regions.

In the near-wake, where two separate wake lobes are present, the deficit profiles are better represented by the superposition of two symmetric Gaussians (Eq. 10).

$$g_2(z) = a\left(\exp\left[-\frac{1}{2}\frac{(z - z_c + \frac{1}{2}z_{sep})^2}{w^2}\right] + \right.$$
$$\left.\exp\left[-\frac{1}{2}\frac{(z - z_c - \frac{1}{2}z_{sep})^2}{w^2}\right]\right) \tag{10}$$

where the center of the wake, $z_c$, is the mean between the centers of the two lobes, separated by a distance $z_{sep}$. The parameters

$a$ and $w$ control the amplitude and width (vertical extent) of the component Gaussian curves. We define the wake maximum deficit to be $\max_z |g_2(z)|$ (not necessarily equal to the parameter $a$ as the separation between the two lobes shrinks). Consistent with the single Gaussian fit, the vertical extent is defined as the full width at which $g_2$ has decayed to $5\%$ of maximum (note



that this definition is not trivially expressed as a constant multiple of the parameter $w$ and is estimated iteratively from the fitted function $g_2$).

We deviate slightly from previous approaches in our formulation of the wake fit. The double Gaussian (Eq. 10) is expressed in terms of the the full wake center, $z_c$, and separation between the lobe centers, $z_{sep}$, in order to more easily place bounds on these parameters. The definitions of the wake magnitude and extent for the double Gaussian fit $g_2$ also differ from previous estimates; we try to keep the definitions intuitively consistent with the behavior of the wake and the single Gaussian at the expense of having simple expressions in terms of the fit parameters $a$ and $w$ (see Appendix A for details).

A nonlinear least-squares fit is performed for both of the Gaussian models (Eqs. 9 and 10) on each of the vertical velocity deficit profiles, moving sequentially downstream of the turbine. To isolate potential wake behavior we use the profile data below 700 m where the deficit falls between -10 % and 100 %. The first profile fit is seeded with hub-height center, $z_c$ and an amplitude of $a = 40$ %. The single Gaussian uses a seed $w = 0.5D$ while the double Gaussian uses $w = 0.3D$ and lobe separation $z_{sep} = 0.5D$. The fitted parameter values are then used to seed the subsequent profile fit downstream. We constrain
the parameter space toward finding physical wake behavior as follows. We restrict $a$ to lie between 0 and 90 % (Porté-Agel et al., 2020)and the center height $z_c$ between 20 and 200 m above the ground. The first profile fit is pinned more closely to the hub height by taking $z_c$ to be within 10 m at 20 m downstream for the LES and within 50 m at the first profile 100 m downstream in observations. For the single Gaussian we confine $w$ between 0 and D$[0, 1.5D]$ and for the double Gaussian we use $w \in [0, 0.5D]$ and $z_{sep} \in [0, 0.75D]$. To encourage the fit to follow a cohesive wake structure, we restrict the parameters so
that the wake center is within 10 m of the previous fit and the height of the wake is within 80 - 120 % of the previous fit. We emphasize that utility in the wake fitting is in providing an accurate quantification of the wake characteristics and choices in parameter ranges are imposed to nudge the Gaussian fit toward the evident wake structures that occur in the datasets.

     Each profile is classified as near wake, far wake, or as having no credible wake detected. Rather than determining the presence of distinct double lobes by a statistical F test, requiring the double Gaussian fit to be significantly better (Aitken and
Lundquist, 2014), we take a different approach based on the idea of the physical convergence of the lobes. When the two Gaussian lobes become so close relative to their width that they have merged, set heuristically here by $z_{sep}/w < 2.2$, it is considered to be part of the far wake (see Appendix A). The wake is considered to have dissipated once the deficit magnitude decays to 5 % of maximum. Any profile further downstream from where a fit has reached $a \leq 5\%$, or $a$ has otherwise reached a minimum, is classified as having no wake with fits considered spurious.

Thus, each profile is given a preliminary, individual classification as (2) near wake with two lobes, (1) far wake with a single lobe, or (0) no wake based on its Gaussian fit. In order to determine coherent wake regions, treating the set of profiles as a whole, the preliminary profile classifications are fitted to a two-tiered logisitic function (Eq. 11) using the nonlinear least-squares method. We use a variation on logistic regression, which is used in binary classification methods (e.g. Spitznagel (2007)), to help create a map between the downstream distance and wake region classifications that adhere to the physical
transition from near to far to no wake (Fig. 4(c)).

$$\ell(x) = 2 - \frac{1}{1 + \exp[-k(x - x_0)]} - \frac{1}{1 + \exp[-k(x - x_1)]} \tag{11}$$





To ensure a sharp transition between two profiles, we take $k = 10$ to be fixed. The parameters $x_0$ and $x_1$ are determined from the functional fit and give the distances at which the wake transitions from near to far and far to dissipated, respectively. Note that the fitting process helps to filter out non-physical variations in the preliminary classification of the individual profiles (e.g., Fig. 4(c) around 220 and 320 m downstream).

The completed wake fit and corresponding wake characteristics (center position, vertical extent, and deficit magnitude) uses the fitted Gaussian model for the determined wake region, i.e., the double Gaussian fit for profiles in the classified near wake and the single Gaussian fit for profiles in the classified far wake.

## 5 Results

### 5.1 LES evaluation of wake observation

In the raw LES flow field, a resonant mountain wave forms over the ridges, much as was observed in the field, and the generalized actuator disk turbine model produces a wake with relatively detailed structure (Fig. 3). The maxima in the drag profile of the blade aerofoils lead to the characteristic dual-lobe structure in the near wake, which merges into the far wake before dissipating. The wake trajectory roughly follows the terrain with the tail of the wake occasionally detaching and lofting a little higher.

In the unwaked transect, the single DTU lidar captures the background flow well, placing the mountain wave and recirculation zones accurately and reproducing all but the most extreme velocity magnitudes. Errors in the horizontal velocity profiles around the reference region for the wake are consistently small ($< 0.1 \mathrm{~m~s}^{-1}$). The accuracy of these measurements is contingent on the smaller spatial and temporal scales in the undisturbed flow compared to those of the lidar probe volume and sweep time, as well as the low beam elevation angles. Because the background flow is captured with high accuracy, it is expected that distortions in the detected wake arise primarily from the measurement of the waked flow itself.

As opposed to flow structures in the freestream transect, smaller spatial scales similar to lidar probe length of 40 m and more rapid variations on the order of minutes pervade the wake and interact more strongly with all of the components of the observation system. Separating the linearly additive stages of the virtual lidar model (Sect. 3.2), we can assess the influence of the resolution of the retrieval points over the wake, the beam projection angle, the RWF probe weighting, and the sweep duration on the measured horizontal velocity profile (Fig. 5). Each of the stages interacts with the wake in a way not seen with the background flow.

Distortions due to the linear interpolation from the retrieval points to the vertical profiles are largest, with values as large as 1 $\mathrm{m~s}^{-1}$, and occur most strongly at the edges of the near wake (Fig. 5(a)). Averaging effects of the RWF over the probe volume reach up to $0.5 \mathrm{~m~s}^{-1}$ and are most pronounced around the upper wake boundary (Fig. 5(c)). Error incurred due the duration of the scan sweep can occasionally be as large as $0.5 \mathrm{~m~s}^{-1}$, peaking around the edges and tail of the wake and during more transient conditions early in the case study (Fig. 5(d)). By design, the elevation angle of the beams is kept small, which proves effective at limiting the error contribution from the projection of the vertical velocity to generally less than $0.2 \mathrm{~m~s}^{-1}$ near the wake, predictably growing with height and the elevation of the beam (Fig. 5(b)).



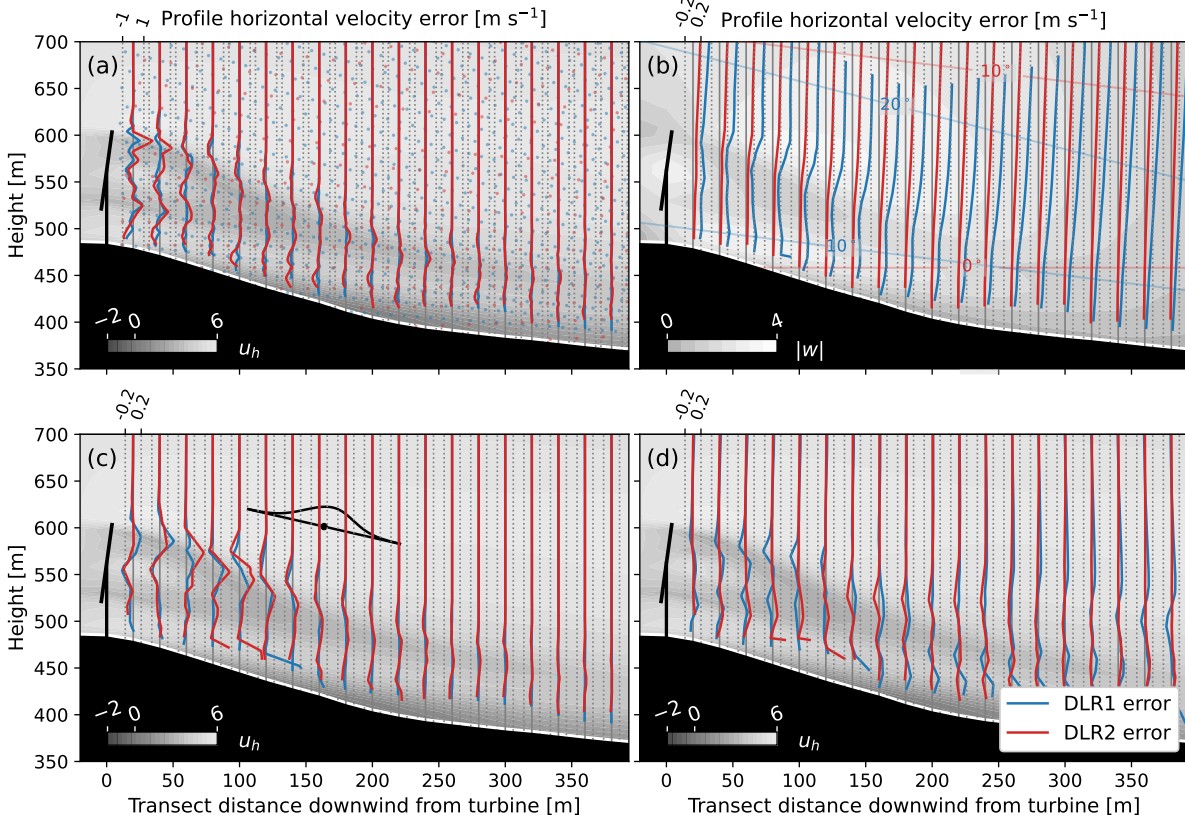

**Figure 5.** Representative snapshot of contributions to error in lidar-derived horizontal velocity profiles due to (a) the position of the lidar retrieval points, (b) contamination by projection of vertical velocity onto the beams, (c) averaging over probe volume through the RWF, and (d) the duration of the scan sweep. Background contours of the LES horizontal (a,c,d) and vertical (b) winds show the wake for reference. Lidar retrieval points, beam elevations, and the RWF are shown for reference in (a), (b), and (c), respectively.

Differences in the positioning and geometry of the two DLR lidar scans provide insight into the variation between altered configurations (reflected in the red and blue lines in Fig. 5). Larger interpolation errors using the DLR2 system evidence the wider spacing between range gates (20 m versus 10 m) and further downstream distance (1000 m versus 1400 m), enlarging the distance between beams at the same angular resolution (1°). We estimate DLR2 to have a density of about 0.002 retrieval points per square meter in the area of the wake compared to 0.006 for DLR1. The positioning of the lidar also consequently

changes the angle of the beams intersecting the wake and therefore the projection and the behavior of the RWF, with the probe volume cutting through and incorporating distinct parts of the wind field. Errors due to the scan duration are marginally larger for DLR1, which takes 77 s to arc across the entire valley to the other ridge, compared to DLR2, which only sweeps to vertical over 51 s. Overall, though differences in the configurations impact components of the error in understandable ways, the geometries are not radically dissimilar, and much of the behavior is echoed across the two systems.

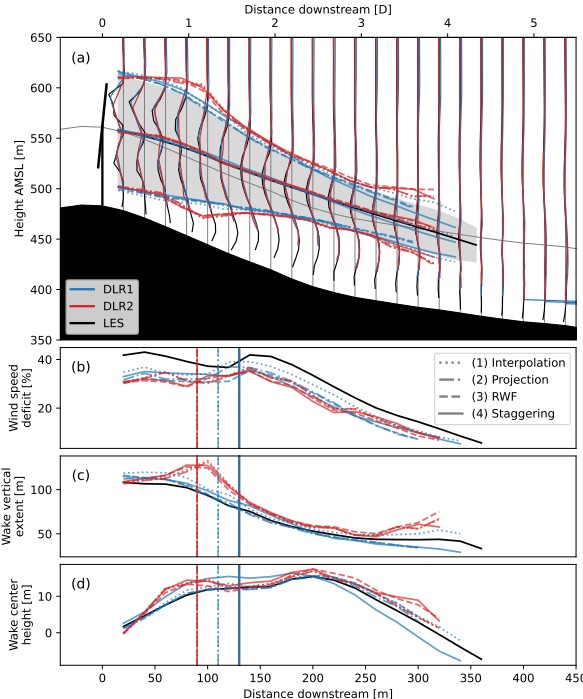

**Figure 6.** (a) Vertical deficit profiles for true LES wake and those measured using the DLR lidars along with the detected wake center line and width. Comparison of detected (b) maximum magnitude of the wake, (c) vertical extent of the wake, and (d) height of wake center relative to terrain-following hub height . Vertical lines show where the fit transitions from a double to single Gaussian, i.e., near to far wake.

Even moderate systematic errors in the velocity field can distort the perceived wake behavior, quantified via the fitted deficit magnitude, vertical extent, and height of the wake center. For a single 5-minute window, we compare the fitted wakes from the time-averaged LES profiles with those from the virtual lidar data (Fig. 6). The wake fits after incorporating each observation stage to the lidar model are compared to elucidate their effect on the retrieved wake. While informative, we caution that the fit operation is nonlinear, so the contribution from the observation stages can no longer be considered additive.

In the overall structure of the wake, we observe that the distinction of the two lobes in the near wake is lost too early in the lidar retrievals (Fig. 6). The early transition exists in the lidar model using only interpolation and may be explained primarily by insufficient resolution of the retrieval points. For DLR1, which has finer resolution than DLR2, RWF averaging also contributes. The DLR1 beams cut through the wake at a steeper angle that may cause more significant smoothing along the vertical axis, blurring the distinction of the lobes.

Using the full lidar model, we show the comparisons of the wake characteristics in aggregate. The virtual instrument retrievals are compared against the LES truth as well as directly against one another. Each of the characteristics are shown in Figs. 7-9 with root-mean-square deviation (RMSD) between the two and metrics for the best-fit least-square lines summarized in the first columns of Table 4.





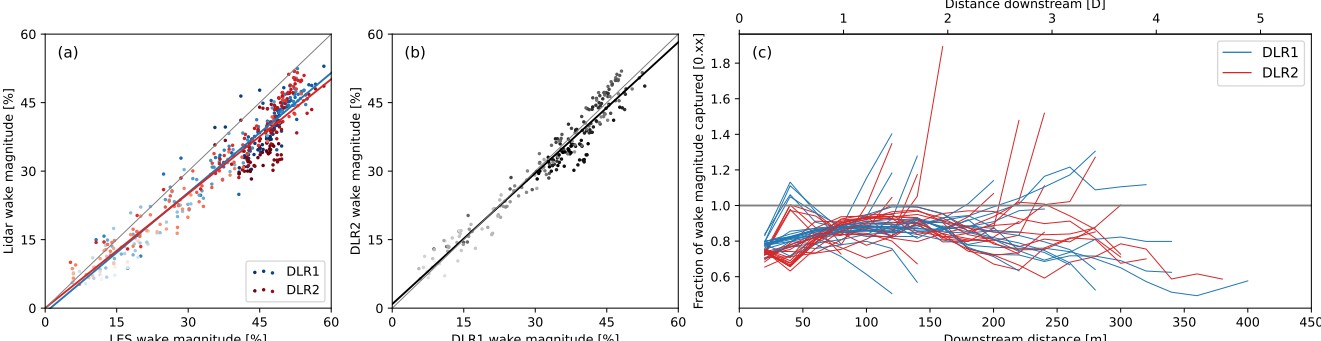

**Figure 7.** Magnitude of the wake deficit with darkest to lightest hues indicating increasing distance from turbine comparing (a) DLR-recovered value plotted against LES truth and (b) DLR instruments plotted against one another with the least-squares best-fit lines. (c) Fraction of peak LES wake deficit captured by the two DLR lidars over wake extent for each 5-minute window.

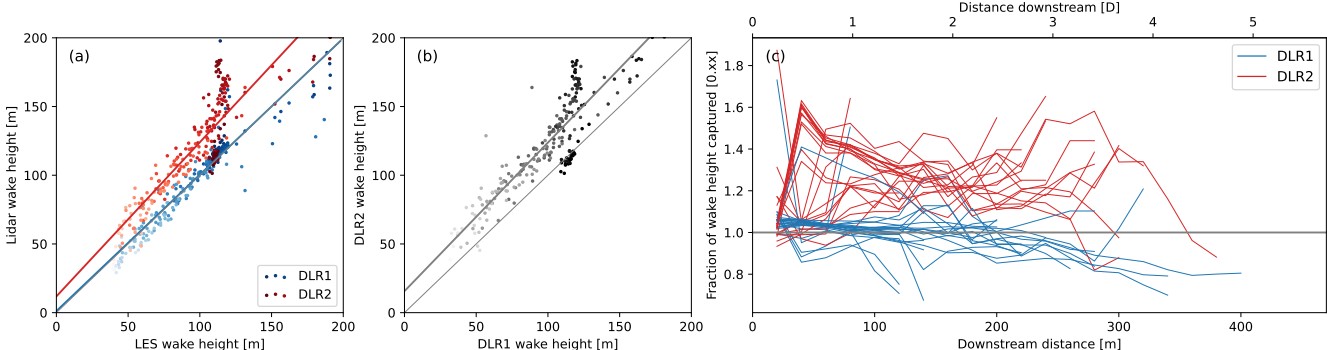

**Figure 8.** Vertical extent of the wake with darkest to lightest hues indicating increasing distance from turbine comparing (a) DLR-recovered value plotted against LES truth and (b) DLR instruments plotted against one another with the least-squares best-fit lines. (c) Fraction of LES wake height captured by the two DLR lidars over wake extent for each 5-minute window.

The peak velocity deficit magnitude is consistently underestimated by both lidars. Figure 7 compares the recovered deficit
against LES in aggregate and tracked over the length of the wake. In the bulk of the wake, only around 0.85 of the maximum wake deficit is reflected in the lidar retrieval (Fig. 7,4). The disaggregation of the stages (Fig. 6(b)) suggests that much of the loss of the peak has already occurred due to the limited resolution of points across the wake and is exacerbated by the further RWF averaging effects over the probe volume. Conversely, elongation and overestimation of the deficit can occur at the tail of the wake, driven by the probe volume (RWF) incorporating slower waked winds from upstream. The DLR2 system on the far
downwind ridge consistently experiences greater deficit loss in the near and middle wake, likely driven by the coarser spatial resolution of the retrieval points identified in the horizontal velocity error. Because both systems underestimate the magnitude, the difference between the two is minor compared to the initial loss in the peak deficit.





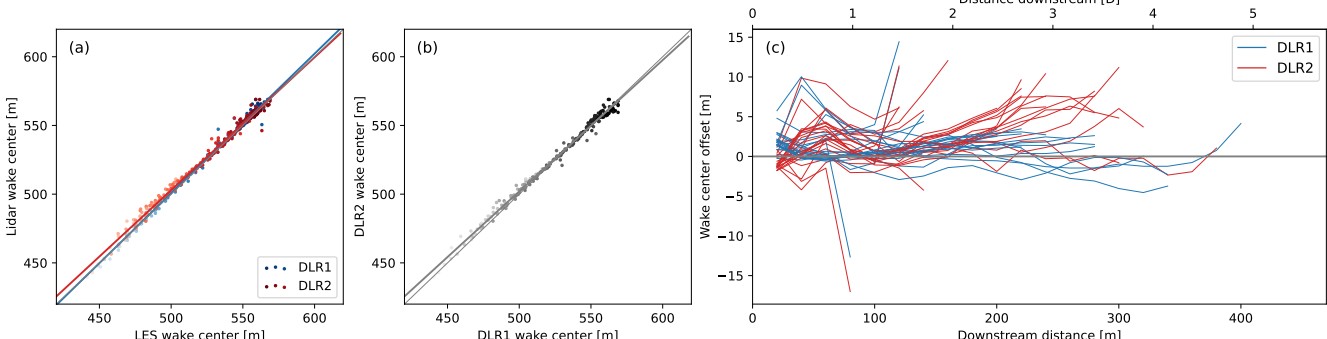

**Figure 9.** Height of the wake center with darkest to lightest hues indicating increasing distance from turbine comparing (a) DLR-recovered value plotted against LES truth and (b) DLR instruments plotted against one another with the least-squares best-fit lines. (c) Difference in the LES wake center height and that captured by the two DLR lidars over wake extent for each 5-minute window.

The vertical extent of the wake in the retrieval is particularly subject to the error noise in the horizontal velocity that occurs around the wake bounds. The wake measured by DLR2 has a height inflated by about 12 m relative to the reference LES, 330 whereas DLR1 is more accurate with errors typically less than 13 m (Fig. 8, Table 4). Interpolation from retrieval points explains most of the difference for DLR2. With DLR1, the distortion is mostly in the tail of the wake and seems to arise once projection effects have been incorporated (Fig. 6(c)).

The trajectory of the wake, tracked via the center of the deficit Gaussian, is the most accurate of the metrics and has the least variability between the two systems (Fig. 9, Table 4). The retrieved position is often offset less than 5 m from the LES reference 335 and rarely exceeds 10 m. Differences between the two systems are minor. DLR2 does suggest a slightly higher trajectory than DLR1 in the tail of the wake and appears to be more sensitive to how the RWF probe volume cut through the wake. The positioning of the wake tail by DLR1 is more significantly affected by the longer scan time, likely due to the more transient behavior of this part of the wake (Fig. 6(d)).

## 5.2 Comparison of observations in field data retrievals

Using data from the three field campaign case studies, spanning 840 total minutes (168 5-minute windows), the same wake data processing successfully yields well-defined wake deficits. For observations, a truth reference is not available, but the independent retrievals from the two DLR lidars may be compared against one another to estimate potential variation due to system configuration. A similar inter-comparison can be done using the virtual instruments so that the approach can simultaneously validate the model as a representation of the behavior of the real data.

We note that the unwaked reference measured by the DTU lidar is used throughout, so the comparison isolates differences in how the two DLR lidar scans collect the wake. In the LES analysis, important distortions, such as the systematic underestimation of the wake deficit magnitude, existed similarly across both systems and are not evident in the inter-comparison between systems.




| | | Simulated Case 3 | | | Observation DLR1 vs DLR2 | | |
|---|---|---|---|---|---|---|---|
| | | LES vs DLR1 | LES vs DLR2 | DLR1 vs DLR2 | Case 1 | Case 2 | Case 3 |
| Magnitude | Fit intercept [%] | -0.95 | 0.03 | 0.88 | -1.54 | 9.46 | 7.74 |
| | Fit slope [–] | 0.87 | 0.83 | 0.96 | 0.79 | 0.80 | 0.86 |
| | Correlation coefficient $r$ [–] | 0.97 | 0.95 | 0.97 | 0.83 | 0.94 | 0.95 |
| | RMSD [%] | 6.84 | 7.69 | 2.76 | 16.53 | 6.64 | 8.63 |
| Height | Fit intercept [m] | 1.16 | 11.83 | 15.65 | 74.52 | 77.10 | 53.61 |
| | Fit slope [–] | 0.99 | 1.12 | 1.08 | 0.76 | 0.55 | 0.67 |
| | Correlation coefficient $r$ [–] | 0.92 | 0.89 | 0.90 | 0.52 | 0.71 | 0.79 |
| | RMSD [m] | 12.41 | 29.86 | 29.26 | 84.10 | 51.70 | 31.95 |
| Center | Fit intercept [m] | -7.34 | 19.69 | 24.36 | -38.65 | 55.58 | 34.23 |
| | Fit slope [–] | 1.02 | 0.97 | 0.96 | 1.05 | 0.89 | 0.94 |
| | Correlation coefficient $r$ [–] | 1.00 | 1.00 | 1.00 | 0.72 | 0.85 | 0.96 |
| | RMSD [m] | 2.54 | 3.81 | 3.14 | 18.60 | 12.06 | 6.25 |

**Table 4.** Root-mean-square deviation (RMSD) and linear least-squares fit metrics for comparisons of wake magnitude, height, and center using virtual and observational retrievals.

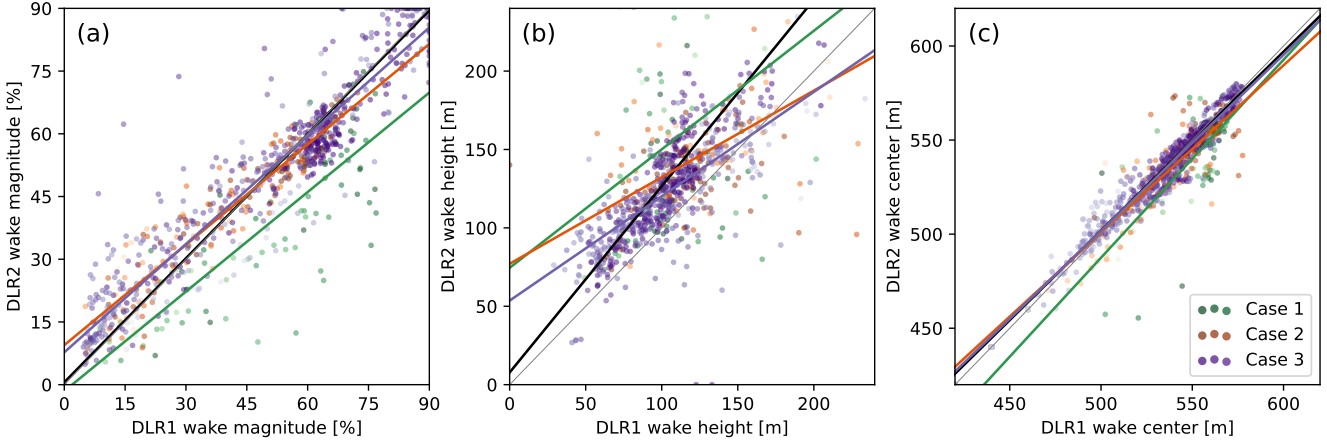

**Figure 10.** Comparison of wake (a) maximum deficit magnitude, (b) vertical extent, and (c) center height position retrieved by DLR1 and DLR2 in the observational case studies. Darkest to lightest hues indicating increasing distance from turbine. The black lines show the best-fit from the virtual instruments in the LES subcase.



Over the course of the selected cases, the observational data reflect a wider range of wind speeds, more transitory conditions,
and greater variety of wake behavior than is covered by the limited simulation period. Evident wake structure in the deficit
profiles occurs in about 86 % of the 5-minute windows. The wake in Case 1 typically follows the terrain line at just above hub
height, but occasionally lofts higher. Velocity deficits are high and the wake often persists far downstream (up to 7 $D$). There
are frequent instances of the wake extent growing with distance, a case neglected in the LES, where the wake almost universally
tapered to dissipation. Case 2 is qualitatively similar to Case 1 though it experiences shorter wakes and weaker initial velocity
deficits. Case 1 features the fastest wind speeds and the wake appears to be completely detached from the terrain variation and
lofts out directly from the hub height.

The comparisons of the observational wake characteristics are noisier due to the nature of the raw data and imperfections
in the wake fitting algorithm. The observational deficit profiles are subject to additional noise from the lidar systems due to
factors like pointing accuracy of the lidar beams, small-scale turbulent behaviors unresolved by the LES, complex aerosol
distributions affecting returns, among other complexities. While the fitting algorithm produces good fits about 90 % of the
time, it does have shortcomings in robustly handling the full range of behavior in the observations. Frequent issues are a lack
of clear truncation point in low-deficit regions and apparent deficit behavior near the surface due to differences in the transect
terrain, which causes the fitted center to track low or the wake signal to be obfuscated.

The wakes retrieved by the two DLR systems show marked differences. Trends comparing wake magnitude, height, and
position largely align with with those of the modeled lidar in the LES (Fig. 10, Table 4). Predictably the best agreements are
with Case 3, which most closely matches the conditions in the LES and has the best quality and quantity of observational data.
The consistency provides confidence in the ability of the model to predict wake measurement behavior in these conditions.

In the deficit magnitude, Cases 2 and 3 reflect predictions from the virtual instruments in which DLR1 reads slightly higher
peak deficits in the near wake and slightly lower in the far wake (Fig. 10(a)). In Case 1, DLR1 consistently retrieves wakes
with a more significantly lower deficit throughout. As in the simulations, the correlations between instruments is strong, though
there the overall variability between the two is higher (6–17 % compared to about 3 % RMSE).

The vertical extent of the wake experiences the most variability between DLR1 and DLR2, even more so than seen in
the LES 10(b)). Because of the large variations, the linear fits are less representative. In Case 3, deviations between the two
retrieved wakes often reach 30 m, a value consistent with the virtual lidar comparison. The other cases see even more dramatic
discrepancies and are marginally correlated.

The trajectory of the wake center in the retrievals agrees well as anticipated. The correlation is high and the linear fits have
slopes close to unity. In Case 3, differences are typically within about 6 m, compared to 3 m with the virtual instruments. Case
2 is similar with a larger spread. Differences are more substantial in Case 1, and we note that for some windows, the differences
in positioning and behavior is evident in the raw deficit profiles.





## 6 Discussion


All of the components of the lidar observation system – the resolution of retrievals in space and time, projection onto the beam, and RWF averaging over the probe volume – interact more strongly with the heterogeneous wake than with the more homogeneous background flow. As the spatial and temporal scales of the wake compete with those of the lidar system and additional vertical velocities are induced in the wake, more significant errors can be introduced into the measurement of the

wake and affect its representation. We caution that the particular distortion behavior is specific to the wake dynamics of any particular time and the geometry of the scan, but insights and generalizable trends can be drawn from the results. Further, the virtual lidar methodology presented here could be applied to other scan types and geometries.

Insufficient spatial resolution of the retrieval points and the smoothing effect of the RWF, which incorporates faster winds in the probe volume, can cause the lidar to miss the largest velocity deficits, underestimating the strength of the wake. Partic-

ularly close to the turbine, these underestimates can be significant. In the far wake, the trend can flip and the lidar sometimes overestimates the magnitude of the wake deficit. Especially near the tail of the wake, the RWF capturing much slower points upwind prevents the measured wake from decaying as quickly as it does in reality. Correspondingly, this effect can cause the measured wake to be longer than in reality when the probe volume still contains waked winds.

The key role of the spatial and temporal resolution on the retrieval is consistent with findings in Doubrawa et al. (2016). In

a more complex setting, and with a fuller lidar model explicitly tracking, e.g., contributions from the RWF, we highlight the potentially dominant contribution to error in horizontal velocity measurements due to under-sampling the wake region. As in Doubrawa et al. (2016) we find that diagnosis of the wake center is typically less sensitive to the coarser sampling than other features of the wake. The retrievals by DLR1 compared to DLR2 illustrates how an increased sampling density can improve the fidelity of the measured wake. The advantage is clear despite larger beam elevation angles and a longer scan sweep. We

emphasize that the DLR scans both included large angular sectors and corresponding long scan durations (51, 77 s); the balance of errors suggests that long scans performing spatially dense sweeps may be a reasonable trade-off.

Echoing Meyer Forsting et al. (2017), our model shows the largest RWF averaging effects occur at near the large gradients at the edges of the wake. The RWF can smooth the edges of the wake, inflating the measured bounds and making it seem taller and longer. While different configurations and technologies (pulsed vs continuous wave) exist that impact the shape and scale

of the RWF, practical considerations in obtaining robust velocity measurements impose hard lower limits on achievable probe volumes. Potential effects of the probe volume and RWF should be taken into account, particularly in areas with large velocity gradients and spatial scales similar to those of the probe.

The observational comparison of independent, concurrent retrievals by the two DLR demonstrates the impacts of differing lidar configurations and underscores the variation that can arise in the measured wakes. The intercomparison of the instruments

roughly coincides with expectations from the virtual instruments and builds confidence in the ability of the model. Some of the additional variability and deviations compared to the simulations, especially in Cases 1 and 2, can likely be ascribed to the wider range of conditions and wake behaviors not reflected in the LES period. For example, the geometry shifts when the wake





lofts or the vertical extent expands rather than shrinking and can produce distortion effects that may differ from the simulated ones. The bulk of the LES period is also remarkably stationary and possible transitory effects may be understated in its results.

## 7 Conclusions

We consider potential distortions in the measurements of a single turbine wake made by scanning lidar at the Perdigão campaign. By employing virtual scanning lidar instruments in a case study simulation, we highlight the distortions that can occur in the retrieval of turbine wakes due to the observation system and compare the behavior with observational data. We focus on RHI scans of a waked transect perpendicular to the parallel ridges at the site by two independent WindCube 200S lidars and use an offset, unwaked transect collected by a third lidar for reference to compute the wake deficit. The center, vertical extent, and deficit magnitude of the wake are extracted using Gaussian fits of the deficit profiles.

Even with a single lidar, the background, unwaked flow is generally well captured (error less than $0.1 \ \mathrm{m \ s^{-1}}$ in reference region) assuming the scanning geometry constrains the beam elevation angles to less than $7°$ from horizontal and vertical velocities less than $3 \ \mathrm{m \ s^{-1}}$. Biases therefore arise from the measurement of the wake itself, particular to the lidar scanning configuration.

Our findings emphasize the overall effectiveness of lidar for wake observations, even with a relatively simple scan configuration. The measurements do largely capture the wake structure over its 4-5 $D$ extent. Compared with the background flow, however, the lidar system interacts more strongly with the scales of the wake and introduce larger errors that can systemically affect the measured behavior of the wake. This analysis suggests where biases may be present and how they arise.

The wake distortion can be largely understood as smoothing effects on the wind field due to sampling resolution and probe-volume averaging. The lidar-retrieved wake dampens the extreme velocity deficits, prematurely loses the distinction of the near wake double lobes, and can blur the wake bounds and extent. For this configuration, both lidars persistently underestimate the maximum velocity deficit by factor of 0.7–0.8. The lidar on the downwind ridge (DLR2) inflates the vertical extent of the wake by a factor of about 1.2; the valley-based lidar (DLR1) is typically more accurate error (<10 m). Both lidar systems most accurately capture the trajectory of the wake center, maintaining errors of typically less than 5 m.

The findings using the virtual LES are reinforced by the observational data. Inter-comparison of DLR1 and DLR2 wake metrics display similar behavior using real and virtual lidar data, suggesting that the model is able to capture how the systems measure the wake and the resulting differences between the two configurations.

The decomposition of the lidar system into stages, enabled by the model, provides insight that can inform potential improvements to scanning strategies. For the Perdigão case, the limitation of the beam elevation angles around the wake prove largely effective at minimizing error due to contamination by projected vertical velocities. Instead, the limited spatial resolution emerges as a leading cause of wake measurement error. We recommend scanning configurations that increase the density of retrieval points over the wake by reducing the spacing between range gates and placement or angular resolution to reduce the distance between beams; the performance of DLR1 compared to DLR2 evidences some of the potential benefits.



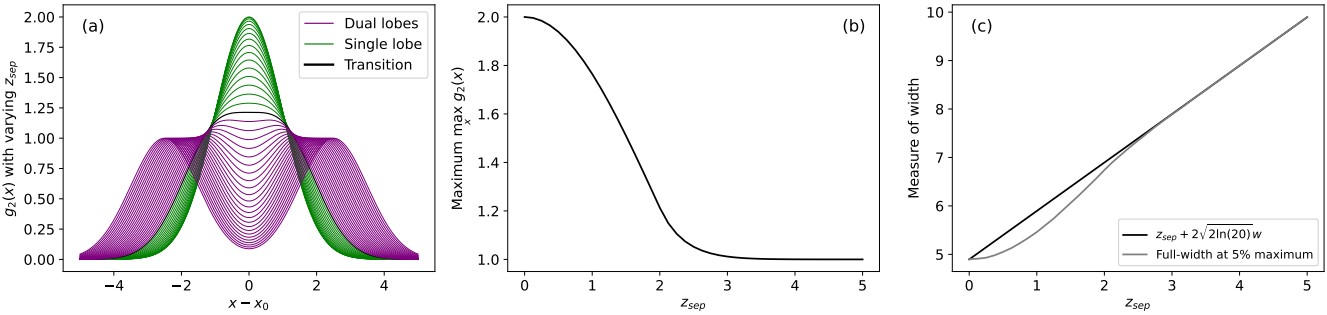

**Figure A1.** (a) Double Gaussian curves, $g_2(x)$, with varying $z_{sep}$ and fixed amplitude $a = 1$ and width $w = 1$ showing the transition of classification of from near (purple) to far (green) wake fits using an empirically picked threshold of $z_{sep} = 2w$ (black). The corresponding (b) maximum and (c) width of the double Gaussian as a function of $z_{sep}$.

Beyond characterizing heterogeneous flows like wind turbines wakes, lidars are also used for assessing heterogeneous flows in urban areas (Newsom et al., 2008; Filioglou et al., 2022), land-water transitions, and internal boundary layers (Krishnamurthy et al., 2023). This virtual lidar tool can help enable quantification of possible errors due to scanning geometries and scanning strategies, to enable optimal field experiment planning and instrument deployment.

*Code and data availability.* The WRF MMC source code is available from https://github.com/a2e-mmc/WRF/tree/mmc_update_v4.3. Vir-
tual lidar code may be found at https://gitlab.com/raro0632/virtual-lidar. Observational lidar data are available as part of the Perdigão campaign database at https://doi.org/10.17616/R31NJMN4. The namelist used for the simulation, the virtual lidar data collected from the LES, and corresponding fitted deficit profiles analyzed here are archived at https://doi.org/10.5281/zenodo.10652098.

## Appendix A: Double Gaussian wake characteristics

Illustration of the behavior of the double Gaussian functions fit to the deficit profiles (Eq. 10, Fig. A1). Representing the
properties of maximum deficit and the vertical extent of the wake cannot always be done as a simple, closed-form function of the input parameters ($a$, $w$, $z_{sep}$). When the two Gaussian lobes overlap more closely, $z_{sep}$ becomes smaller relative to the widths of the individual Gaussian lobes, $w$, they add in a more complex way.

  To be consistent with the single Gaussian properties and to accurately reflect the physical behavior we care about, we directly estimate the maximum value of $g_2$ and where it decays to 5 % of its maximum to determine the amplitude and width. These
estimates are done with evaluation of the function at many points rather than a direct calculation with the input parameters.

  The transition from near to far wake is determined by the double Gaussian fit. The merging of the individual lobes is reflected in a smaller separation relative to the width of each lobe so that they are no longer distinct. An empirically picked threshold of $z_{sep} = 2w$ is used as a cutoff (Figure A1(a)).





*Author contributions.* **Rachel Robey:** methodology, software, analysis and investigation, writing and editing. **Julie K. Lundquist:** conceptualization, methodology, writing and editing.

*Competing interests.* At least one of the (co-)authors is a member of the editorial board of Wind Energy Science.

*Disclaimer.* This work was authored in part by the National Renewable Energy Laboratory, operated by Alliance for Sustainable Energy, LLC, for the U.S. Department of Energy (DOE) under Contract No. DE-AC36-08GO28308. Funding provided by the U.S. Department of Energy Office of Energy Efficiency and Renewable Energy Wind Energy Technologies Office. The views expressed in the article do not necessarily represent the views of the DOE or the U.S. Government. The U.S. Government retains and the publisher, by accepting the article for publication, acknowledges that the U.S. Government retains a nonexclusive paid-up, irrevocable, worldwide license to publish or reproduce the published form of this work, or allow others to do so, for U.S. Government purposes.

*Acknowledgements.* Thanks to Adam Wise, who did the original Perdigão simulations in WRF, for his help matching his configuration and making the necessary updates to run with WRF 4.3. Thanks to Nathan Agarwal for processing and filtering tower data for identification and characterization of conditions for case studies within the field campaign.

This material is based upon work supported by the U.S. Department of Energy, Office of Science, Office of Advanced Scientific Computing Research, Department of Energy Computational Science Graduate Fellowship under Award Number DE-SC0021110.

This research has been supported by the US National Science Foundation (grant no. AGS-1565498).

We would like to acknowledge high-performance-computing support from Cheyenne (doi:10.5065/D6RX99HX) provided by National Center for Atmospheric Research's Computational and Information Systems Laboratory, sponsored by the National Science Foundation.



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
