# Peer review of "Influences of lidar scanning parameters on wind turbine wake retrievals in complex terrain"

_Wind Energy Science, 2024_

## Referee Comment (RC2)

**Review of "Influences of lidar scanning parameters on wind turbine wake retrievals in complex terrain"**

**Authors: Rachel Robey and Julie K. Lundquist**

This manuscript employs large-eddy simulation to simulate the scanning Doppler LiDAR-based measurements of wind turbine wakes. The study considers three LiDAR system installed in a complex terrain of Perdigão field measurement site and investigate how scanning parameters influence the wake observation. Since LiDAR-based wind field measurements is receiving a lot of attention in wind energy, from early planning stage for wind resource assessments to turbine control during the operation phase, studies like the one presented in the current manuscript is important to understand the accuracy of LiDAR technology. However, the authors need to revise and improve the manuscript significantly before it can be accepted for publication. Discussions are confusing and at many places incomplete. It is not always possible to understand which figure you are explaining in the particular paragraph. The authors are asked to address the following comments in the revised submission.

**Specific comments:**

1. Pg1, line 18: "Scanning lidars . . . "
   Reference(s) is required

2. In the Introduction (1st and 2nd paragraphs), you have mentioned about many earlier studies, but have not provided any information about their findings and contributions. Furthermore, you also need to mention what was missing in those studies so that you can link them with the objectives of the current manuscript.

3. Pg. 2, line 27 through 31: The name of Perdigão field campaign site appears abruptly over here. Since you are performing LES for this site you may move this part to the last or the 2nd last paragraph.

4. Description of figures in general is confusing. It is not clear whether you are describing the results presented in specific figure or simply providing general discussion. Please start with more soecific statement like "Figure . . . shows...", and then describe the figure

5. Figure 1: It will be helpful if you can add a table summarizing instruments and turbine.

6. Pg. 3, line 62: ". . . from the upwnd ridge . . . "
   If you wan to add upwind ridge over here, you should add and arrow indicating dominant wind direction in Fig. 1.

7. Fig. 2: Why do you have only one data per hour in this figure? Please show 10-minute or 5-minute mean wind and stability data.

8. Eq. (1): Please describe how do you obtain $u_*$, $\Theta_0$ and $\overline{w'\theta'}$.

9. Pg. 4, line 75 and Fig. 2: "The selected cases . . . neutral to stable"
   Is there a reason for selecting neutral to stable case? You also need to discuss why stability differs significantly with height in Fig. 2.

10. Pg. 5, line 81 through 83:
    This sentence is not clear. What do you mean by "instrument 85 to 89"?

11. Pg. 7: Table 3: I do not think readers will understand parameters in this table and what do they stand for. You need to provide more information either in the text of in the same table.

12. Pg. 7, line 106 through 116: You need to provide more information about the LES setup. Are you using periodic boundary conditions in the horizontal direction? If so, do you have any special way of generating inflow fields? How have you implemented stability? Do you define initial potential temperature profile?

13. Pg. 8, line 133: "linear barycentric interpolation"
    You can describe this for your case in an appendix.

14. Pg. 8, Eq. (3): What are $s$ and $ds$?

15. Pg. 9, line 179 to 186: I am not sure if elevation angle of $\phi = 20°$ is sufficiently small to exclude vertical velocity component. This is particularly the case for complex terrain. You should discuss the challenges of including $w$ while computing velocities from $v_r$. Furthermore, in the result section you should show the error incurred due to this assumption. That should be possible, since you have access to 3D LES data.

16. Pg. 10, line 195: "...any sweep with time stamps inside the window are included."
    What does this phrase mean? Please make this statement clear.

17. Pg. 10, line 211: "Maximum drag on ...the blade tip, ..."
    From Fig. 4(a), it seems maximum velocity deficit occurs around the middle of the blade. This does not agree with your statement.

18. Pg 13, line 271 through 281: Are you trying to discuss the results in Fig. 5 over here?

19. Pg. 13, line 292: "By design, ..."
    You cannot make such a statement about elevation angle, until you compare results from two or more elevation angles?

20. Figure 6 (a): The wake width drops significantly downstream from 3D. This seems unnatural to me. Also, two lobes disappear too early ($\leq$1.5D) compared to 4 to 5D in many wind tunnel studies. Is it solely due to "insufficient resolution of retrieval points" as stated by the author OR complex terrain is also a contributing factor? This should be clarified in the manuscript.

21. Pg. 16, line 319: What is "peak velocity deficit magnitude" in Fig. 7? Please define it...if necessary using an equation.

22. Fig. 7: What is the definition of the "magnitude of the wake deficit". Please add its mathematical definition (or description) in the text.

23. How can you relate simulation results (Fig. 7, 8, 9) against the observations in Fig. 10? This is not clear from the discussion in section 5.1 and 5.2.

24. Pg. 21, line 439 through 441: I do not think you have verified these statements regarding elevation angle. For example, you could have done scan at higher elevation angle and compared the results against 20° elevation limit you have set.

25. Pg. 22, line 445 through 448: You can move the literature review to Introduction. I do not see the point in putting this paragraph over here.

**Minor comments and corrections:**

1. Pg2, line 48: Section 3 is missing.

2. Table 1: Date format ins the 1st and the 2nd rows should be same. Also, add standard deviation of wind direction for all three cases.

3. Pg. 4, line 67: "three towers ..."
   I can only see two towers in Fig. 1. Where is the third tower.

4. Pg. 5, line 92: Do you mean CNR $> -24$dB?

5. Pg. 15, line 305 through 314: Should be a single paragraph.

6. Pg. 20, line 403: "...at near..."
   Remove at

---

## Author Comment (AC1)

**General comments from the author on the revision**

We appreciate the editor's and the reviewers' time and effort to provide careful and thoughtful feedback and reviews of our manuscript. We have made edits to the manuscript to address concerns and issues identified by the reviewers. Based on the responses from both reviewers, we have reworked the introduction to better summarize findings in previous studies and contextualize the novelty of our contributions. The changes also include updates to the following figures:

- Fig. 01 to include tower 13 and the direction of incoming winds
- Fig. 02 to remove the 100-m stability metrics and to use higher-frequency (1-min) wind speed and direction
- Fig. 03 to make the "Elevation angle" label black, consistent with the rest of the labels
- Fig. 06 to add 'LES truth' to the legend in panel (b)
- Fig. A1 to update the legend in panel (c) to use 'σ' in place of 'w', consistent with changes in the text

Below are our point-by-point responses to the reviewer comments which appear in black, while our responses appear in blue. Line numbers in our responses refer to the updated manuscript.

**Referee #1**

**General Comments**

The manuscript WES-2024-18 investigates the impact of the operating configuration and of the technical specifications of a scanning wind lidar on the measurement of complex wind flow. The focus is on quantifying biases in wind speed observations attributed to the speed and geometry of the scanning pattern used, as well as to the probe length and the spatial resolution of a wind lidar. The investigation takes place using wind lidar observations, acquired in the context of the Perdigão experiment, which are coupled with a WRF-LES simulation. The article is well structured and written, has a good introduction of previous related studies and presents in a thorough way the analysis and the results of the study. I would like to congratulate the authors for that. However, it is difficult to identify what is the novelty of the study.

Please find below my specific comments and suggestions for minor corrections that intent to clarify the work presented in this manuscript. You will notice that most of my comments are minor. However, as I already mentioned the authors should emphasize on the new findings that this study presents. Furthermore, I have two minor comments that concern the whole manuscript. The authors use the terms "spatially resolved measurements" and the verb "interacts" (e.g. "the observation system interacts with the smaller spatial…") in a way that can be confusing according to my opinion. I can understand the term "spatially distributed measurements", but there is nothing resolved in the lidar measurements. The measurements provide spatially distributed observations of the radial speeds. Furthermore, there is no "interaction" between a wind lidar and the atmospheric wind since the wind characteristics are not distorted due to the operation of the wind lidar. I think that these two points should be clarified throughout the manuscript.

Our thanks to the reviewer for their time and thoughtful comments including identification of a lack of clarity regarding the novel contributions of the study and in the nuances of wording choices.

We would like to first address the major concern about clarifying the novelty of the study. In comparison to the previous cited studies, Meyer Forsting (2017) and Doubrawa (2016) in particular, there are some key differences in setting and analysis approach in our study:

- We treat a real setting in a complex case study from the Perdigão campaign, using both model and the observations, with important topological variation and nested boundary conditions that allows for the development of complex flow conditions including the mountain wave and valley recirculations. The cited studies focus on more idealized conditions and flat sites, with the models using prescribed and periodic boundary conditions. (Doubrawa (2016) does consider observational measurements at a flat site in conjunction with the modeled lidar measurements).
- As we note, the scan configuration and geometry is important to the behavior of the retrieval error. The lidar in this Perdigão case study are sited downwind looking back toward the turbine rather than the colocated/nacelle based lidar in the cited studies.
- Novel analysis identifies the linear additivity of the error contributions from different portions of the observation system to disaggregate and attribute contributions to error in the measured wake due to multiple sources (spatial/temporal resolution of the measurements, averaging over the probe volume / RWF, and beam pointing / contamination of the horizontal velocity due to projection of vertical velocities). Doubrawa (2016) focuses on the temporal/spatial resolution of the measurements using 3D, stacked-sector scans which are much larger and take much longer (12 min) and omit the effects of the probe volume averaging. Meyer Forsting (2017) conversely focuses on the volume averaging and doesn't appear to resolve the specific scan pattern or duration. In both cases, the behavior due to velocity projection onto the beam might be expected to differ in complex terrain compared to flat conditions. In our approach, we contextualize and compare of the contributions to error from each of these effects.

In addition to the new approaches in the analysis, we believe the study documents and contributes to larger discussions about how the error may behave across different settings and scan approaches. In the manuscript, we have edited the introduction (lines 38-55) and discussion (lines 410-412 and 417-418) to give greater detail about the related studies and better highlight how our approach and findings relate to these specifics.

In the wording choice, we have adopted the suggestion of "spatially distributed measurements" in place of "spatially resolved measurements" throughout. While we had hoped to use "interact" to convey the idea of the coincidence of the particular character of the flow and the scales/properties of the lidar system together producing error behavior, we do not want to mistakenly imply that there s an actual distortion of the flow due to the instrument – we defer to the suggestion of removing this wording and have replaced it with clearer language in which the lidar "reacts or responds to" the flow or that the system is "more prone to error" acting on the waked vs unwaked flow.

**Specific comments**

1. Pg.5 Line 81. What does it mean that in the archive the DLR lidars changed number and why is it relevant for the study.

   The DLR instrument on the downwind ridge had to be swapped during the campaign for technical reasons, so that the instrument with ID 85 was replaced with the instrument with ID

no. 89; the relevance of including this was only in providing documentation for reproducibility about which instrument was at this position when accessing archived data.

Since it is causing confusion and not directly relevant to the methods, we have removed this sentence so that it doesn't disrupt the discussion. We have instead added a note to the data availability statement: "Observational lidar data are available as part of the Perdigão campaign database at https://doi.org/10.17616/R31NJMN4 (note that the lidar designated by DLR2 was swapped on 3 June 19:20 UTC from instrument number 85 to 89 in the archive)." We find this note to be important for others who might use this dataset.

2. Pg.8 Lines 132 - 134. Have the authors tried also other types of interpolations? And how sensitive are the conclusions of the study about the contribution to the error in the simulated lidar measurements due to the interpolation method used?

The interpolation from the wind field is an expensive part of the virtual lidar and we believed it out of scope for this study to try other types of interpolation. Assuming the LES is well enough resolved, we would not anticipate large differences due to the choice of interpolation method.

Barycentric linear interpolation is a common generalization of linear interpolation to multiple dimensions for irregular points (the method is what underlies scipy's LinearNDIterpolator). The idea is a simple, stable approach which is a local, piecewise linear interpolation over the triangle/tetrahedron of nearest points. Given that the grid may be somewhat irregular in the vertical between columns (WRF LES uses a terrain-following pressure coordinate), we chose this approach to represent the resolved flow field using the closest points and without smearing out details.

We have added to line 147 references to Scipy documentation (Virtanen et al., 2020) and a good summary of the method in (Amidror et al. 2002)

3. Pg.13 Line 288-289. What is the interpolated 2D wind field compared with? How are the "distortions" calculated?

The distortions are computed by subtracting the true, underlying LES velocity from the velocity seen by the lidar (after the LES flow has been filtered through the observation system), i.e.,
    error = u_lidar - u_LES
Here we isolate just the error due to the spatial resolution of the measurement points (which we called the "linear interpolation from the retrieval points to the vertical profiles" in this line). The true, well resolved LES flow field is subtracted from the flow field seen just by the lower-resolution virtual lidar measurement points (ignoring line-of-sight projection, probe volume averaging, and temporal advancement of the beam).

To try to clarify this in the text we have added the statement "Errors are computed with respect to the "true", underlying LES flow field." to the preceding paragraph and edited

"Distortions due to the linear interpolation from the retrieval points to the vertical profiles are largest, …"

to "The largest contribution to the error in the horizontal velocity profile is due to the spatial resolution of the lidar measurement points, …"

4.  Pg. 15 Figure 6. What does the solid black line represent?

We regret there wasn't more clarity about what was being shown here. The solid black line represents the 'true' values from the fitted, underlying LES flow; we have corrected the figure to explicitly add this label to the legend.

5.  Pg. 16 Lines 323 – 324. Can the authors explain a bit more why they expect to see an elongation of the wake due to the probe volume? Isn't this dependent on the magnitude of the wind speed gradient within the probe volume?

Given any measurement that reflects an average over a nonnegligible probe volume, target points outside the wake can incorporate points from inside the wake. If the probe volume incorporates both waked and non-waked data points (sketched below), that measurement point will seem to be waked because the averaging over the probe volume makes the velocity slower than in actuality. This leads to "smearing" or "blurring" of the wake, including in the length. The degree to which this occurs will indeed depend on the magnitude of the wind speed gradient and length/weighting of the probe volume, but even without knowing these details we might expect the possibility of this blurring/elongation effect.

We have expanded the statement (line 339)

"Conversely, elongation and overestimation of the deficit can occur at the tail of the wake, driven by the probe volume (RWF) incorporating slower waked winds from upstream."

to give a little more detail

"Conversely, elongation and overestimation of the deficit can occur at the tail of the wake, driven by the probe volume (RWF); measurements made at target points beyond the wake can incorporate slower waked winds from upstream, causing unwaked points to appear to be waked and blurring the wake bounds."

[Figure]

**Minor corrections:**

Pg.2 Line 33. I suggest replacing "…to ensure robust returns,…" with "to ensure adequate backscattering signals…" or something similar.

Changed "sufficient air quality to ensure robust returns" to "sufficient air quality to ensure adequate backscatter signals"

Pg.2 Line 43. I suggest replacing "and virtual instrument…" with "and a virtual lidar…" Done.

Pg.2 Figure 1. Is it possible to add the location of the tower tw13, since observations from that tower are presented in Fig. 2. Yes; the figure has been fixed.

Pg.4 Line 70. Add the units of the gravitational acceleration. Corrected and we've added units for the other values here for consistency.

Pg.4 Line 77. It is written: "the positions and scan configurations of the three Danish …" but as far as I understand only one wind lidar belongs to DTU. Can you please clarify this part.

Your understanding is correct and this was poorly worded referring to 3 lidar = 1 DTU and 2 DLR lidar. The sentence has been corrected to say "scan configurations of the Danish Technical University (DTU) lidar and two German Aerospace Center (DLR) lidars analyzed in this study"

Pg.5 Line 85. The word "cuts" could be replaced by the word "planes". Done.

Pg. 7 Line 122. Replace "off of" with "by" Corrected.

Pg. 8 Line 145. Add "the" before "WindCube" Corrected.

Pg. 10 Line 217. The symbol "w" in Eq. 9 is also used to denote the vertical component of the wind vector. I suggest replacing it with something else to avoid confusion.

> Thanks for catching the repetition in notation and potential confusion. We've replaced "w" with "σ" throughout where it was used to represent the wake parameter as in Eq. 9.

Pg. 12 Line 231. Delete one "the" from "… in terms of the the …" Done.

Pg. 13 Line 282. Is the probe length equal to 40 m or 44 m (as reported in Table 3)?

> The 40 m was used as a rough figure; this has been corrected to the more exact 44 m.

Pg. 16 Line 321. Is the (Fig. 7,4) correct? Thanks for catching, this should be (Fig. 7, Table 4). Corrected.

Pg. 19 Line 365. Delete one "with" from "align with with" Corrected.

References: Please check the information of the references is complete. Also, the Menke et al 2019b is reported twice. Thanks for catching; corrected.

**Referee #2**

**General comments:**

This manuscript employs large-eddy simulation to simulate the scanning Doppler LiDAR-based measurements of wind turbine wakes. The study considers three LiDAR system installed in a complex terrain of Perdig˜ao field measurement site and investigate how scanning parameters influence the wake observation. Since LiDAR-based wind field measurements is receiving a lot of attention in wind energy, from early planning stage for wind resource assessments to turbine control during the operation phase, studies like the one presented in the current manuscript is important to understand the accuracy of LiDAR technology. However, the authors need to revise and improve the manuscript significantly before it can be accepted for publication. Discussions are confusing and at many places incomplete. It is not always possible to understand which figure you are explaining in the particular paragraph. The authors are asked to address the following comments in the revised submission.

We would like to thank the reviewer for their time and comments to hopefully make our presentation as clear as possible. We appreciate catching some important errors and typos which we are grateful to have corrected for the updated manuscript.

**Specific comments:**

1. Pg1, line 18: "Scanning lidars . . . " Reference(s) is required

We've added a few references here: "Scanning lidars are increasingly employed in the collection of spatially resolved measurements of wind turbine wakes (Menke et al., 2019b; Bodini et al., 2017b; Moriarty et al., 2020), …"

2.  In the Introduction (1st and 2nd paragraphs), you have mentioned about many earlier studies, but have not provided any information about their findings and contributions. Furthermore, you also need to mention what was missing in those studies so that you can link them with the objectives of the current manuscript.

We have edited the introduction (lines 38-55) and discussion (lines 410-412 and 417-418) to give greater detail about the related, previous studies and better highlight key differences in setting and analysis approach in our study and its motivation.

3.  Pg. 2, line 27 through 31: The name of Perdigao field campaign site appears abruptly over here. Since you are performing LES for this site you may move this part to the last or the 2nd last paragraph.

We have refined the segue into discussion of Perdigao from:

"The Perdigão field campaign was a seminal study of the behavior and measurement of turbine wakes in complex terrain (Fernando et al., 2019); the site features parallel double ridges with a single 2 MW turbine, the wake of which was captured through an extensive array of scanning lidar (Fig. 1)."

to (lines 26-30)

"More recent studies have explored the variability of wakes in complex terrain. The Perdigão field campaign (Fernando et al., 2019) was a seminal study at a site featuring parallel double ridges and a single 2 MW turbine, the wake of which was captured through an extensive array of scanning lidar (Fig. 1)."

4.  Description of figures in general is confusing. It is not clear whether you are describing the results presented in specific figure or simply providing general discussion. Please start with more specific statement like "Figure . . . shows...", and then describe the figure

We have expanded the introduction to the results figures as follows:

"Separating the linearly additive stages of the virtual lidar model (Sect. 3.2), we can assess the influence of the resolution of the retrieval points over the wake, the beam projection angle, the RWF probe weighting, and the sweep duration on the measured horizontal velocity profile (Fig.5)"

→ "... horizontal velocity profile. Figure 5 shows the linear contribution to the total error in the horizontal velocity profiles downwind of the turbine due to each of these sources."

"Using the full lidar model, we show the comparisons of the wake characteristics in aggregate. The virtual instrument retrievals are compared against the LES truth as well as directly against one another. Each of the characteristics are shown in Figs. 7-10 with

root-mean-square deviation (RMSD) between the two and metrics for the best-fit least-square lines summarized in the first columns of Table 4."

→ "Using the full lidar model, we show the comparisons of the wake characteristics in aggregate. The wake deficit magnitude, vertical extent, and center position are shown in Figs. 7-10, plotting the lidar retrieved values against the LES truth as well as directly against one another and showing the error in the retrievals over the the wake extent. The root-mean-square deviation (RMSD) between the true and measured wakes and metrics for the best-fit least-square lines summarized in the first columns of Table 4."

"For observations, a truth reference is not available, but the independent retrievals from the two DLR lidars may be compared against one another to estimate potential variation due to system configuration (Fig. 10). A similar inter-comparison can be done using the virtual instruments (as shown in panel (b) of Fig. 7-9) so that the approach can simultaneously validate the model as a representation of the behavior of the real data."

We have made an effort to introduce the figures and have descriptive captions and all out the figures and specific panels in the text where they are discussed or supporting a particular statement.

5. Figure 1: It will be helpful if you can add a table summarizing instruments and turbine.

   Table 2 summarizes the position and sweep parameters for the scanning lidar instruments. Because there is only one turbine, we don't think adding its information to a new table is an efficient use of space.

6. Pg. 3, line 62: ". . . from the upwind ridge . . . " If you want to add upwind ridge over here, you should add and arrow indicating dominant wind direction in Fig. 1. An arrow indicating the wind has been added.

7. Fig. 2: Why do you have only one data per hour in this figure? Please show 10-minute or 5-minute mean wind and stability data.

   The data in Figure 2 are reported every 30 minutes, as noted in the caption. For the wind speed and direction data, we have updated the plot to show higher-frequency, 1-minute averages.

   Because we are considering atmospheric stability as calculated using fluxes, and a 30-minute Reynolds averaging period is recommended for flux calculations in these stability conditions (e.g. Foken & Wichura 1996), we have chosen to keep the 30-minute averaging period for the stability metrics.

8. Eq. (1): Please describe how do you obtain $u_*$, $\Theta 0$ and $\overline{w'\theta'}$.

   We have added a description of how these are computed: "Heat and momentum fluxes are computed with a standard eddy-covariance approach from 20-Hz sonic anemometer and 1-Hz air temperature sensor data over a 30-minute averaging window."

9. Pg. 4, line 75 and Fig. 2: "The selected cases . . . neutral to stable" Is there a reason for selecting neutral to stable case? You also need to discuss why stability differs significantly with height in Fig. 2.

Yes, there is a reason for focusing on stable cases: we are observing wind turbine wakes which are known to be more coherent and detectable in stable conditions (Bodini et al. 2017). We agree that it is potentially confusing to include stability measured from fluxes at different levels and so the revised figure uses only the standard Obukhov length as measured at the lowest altitude available, 10m.

To better reflect these ideas, we have also modified the text from

"Over the complex topography of the site, significant variations in the stability parameter can occur, reflected by the variation in the values reported at the different towers and heights. The selected cases highlighted here are mostly neutral to stable (Fig. 2)."

to

"Over the complex topography of the site, significant variations in the stability parameter can occur, reflected by the variation in the values reported at the different towers. The selected cases are mostly neutral to stable (Fig. 2), conditions under which the wake is expected to be more coherent and detectable (Bodini et al. 2017)."

10. Pg. 5, line 81 through 83: This sentence is not clear. What do you mean by "instrument 85 to 89"?

This sentence was also called out by reviewer 1 as being unclear. As in that response:

The DLR instrument on the downwind ridge had to be swapped during the campaign for technical reasons, so that the instrument with ID 85 was replaced with the instrument with ID no. 89; the relevance of including this was only in providing documentation for reproducibility about which instrument was at this position when accessing archived data. Since it is causing confusion and not directly relevant to the methods, we have removed this sentence so that it doesn't disrupt the discussion. We have instead added a note to the data availability statement: "Observational lidar data are available as part of the Perdigão campaign database at https://doi.org/10.17616/R31NJMN4 (note that the lidar designated by DLR2 was swapped on 3 June 19:20 UTC from instrument number 85 to 89 in the archive)."

11. Pg. 7: Table 3: I do not think readers will understand parameters in this table and what do they stand for. You need to provide more information either in the text of in the same table.

Other papers investigating lidar behavior in the same level of detail that we do also incorporate similar tables, see e.g. Wildmann et al. 2018 Table 1, Stawiarski et al 2013 Table 2, Cheynet et al 2017 Table 2. It is not possible to understand the detail of lidar operations and behavior of the probe volume for a given system without these parameters. The parameters are defined in the discussion of the range gate weighting function (RWF) in Section 3.2. Readers are welcome to skim through these, but we do believe some will find

them important for understanding how the system settings gave rise to the probe volume and weighting.

12. Pg. 7, line 106 through 116: You need to provide more information about the LES setup. Are you using periodic boundary conditions in the horizontal direction? If so, do you have any special way of generating inflow fields? How have you implemented stability? Do you define initial potential temperature profile?

In the description of the LES, we state that "As in Wise et al. (2022), five domains are used to nest down from mesoscale simulations" – this means that real boundary conditions, not periodic boundary conditions, are used, and stability is driven by the nested simulations. More information on mesoscale-microscale modeling can be found in Haupt et al. 2019 (https://journals.ametsoc.org/view/journals/bams/100/12/bams-d-18-0033.1.xml) and Haupt et al. 2022 (https://wes.copernicus.org/articles/8/1251/2023/wes-8-1251-2023.html).

To make the approach clearer for a wider audience, we have added the following sentence (after "Wise") referring interested readers to mesoscale-microscale modeling approaches and these references.

"This mesoscale-microscale modeling approach allows realistic inflow and forcing in the LES, driven by coarser models of the larger weather systems (Haupt et al., 2019, 2023)."

13. Pg. 8, line 133: "linear barycentric interpolation" You can describe this for your case in an appendix.

Barycentric linear interpolation is an established generalization of linear interpolation to multiple dimensions for irregular points (the method is what underlies scipy's LinearNDIterpolator). The idea is to use a simple, stable approach which is a local, piecewise linear interpolation over the triangle/tetrahedron of nearest points. There are not unique considerations for its application to our case, so rather than include an appendix we have listed a few references here and added citations to the text for the interested reader.

References:

Virtanen, P. *et al.* SciPy 1.0: Fundamental algorithms for scientific computing in python.

*Nature Methods* **17**, 261–272 (2020).

Amidror, I. Scattered data interpolation methods for electronic imaging systems: A

survey. *Journal of Electronic Imaging* **11**, (2002).

(Hughes, J. F. *et al. Computer Graphics: Principles and Practice*. (Addison-Wesley,

Upper Saddle River, NJ, 2013). (Chap 9))

14. Pg. 8, Eq. (3): What are s and ds?

Here, s and ds are simply dummy variables of integration radially along the beam; we believe the description of the integral above the equation as "the convolution of the projected wind velocities along the beam with a range-gate weighting function" is a sufficient description.

15. Pg. 9, line 179 to 186: I am not sure if elevation angle of φ = 20∘ is sufficiently small to exclude vertical velocity component. This is particularly the case for complex terrain. You should discuss the challenges of including w while computing velocities from vr. Furthermore, in the result section you should show the error incurred due to this assumption. That should be possible, since you have access to 3D LES data.

As you suggest, we do in fact explicitly treat the error incurred by violations of this assumption, i.e. due to non-negligible projections of the vertical velocity, by leveraging the underlying 3D LES data. We show and discuss this error in the results in Fig. 5(b) (reproduced below) and lines 309-311.

[Figure]

(Fig 5b) "contamination by projection of vertical velocity onto the beams"

To emphasize that in this part of the text we are only introducing the idea of constraining the projection error by limiting elevation angle, and to hopefully better signal that we are not yet assessing the error due to the assumption and that 20∘ reflects an idea of the values used and not an arbitrary suggested angle, we have edited the original text:

"To estimate the horizontal velocity from radial velocities, $v_r$ , the projection of the vertical velocity is assumed to be negligible, meaning that the radial velocities are purely a projection of the horizontal velocity. Equation 7 solves for the horizontal velocity under this assumption.

$$u_{h,lidar} = v_r \cos \varphi = u_h + w \tan \varphi \quad (7)$$

where φ is the beam elevation angle. The assumption is robust so long as uh ≫ w tan φ. Significant vertical velocities do occur given the complex terrain and wake, but the elevation angles of the beams scanning the upwind ridge in the area of the wake are kept small, φ < 20◦ (tan 20◦ ≈ 0.36), constraining the potential contamination from vertical velocities when relying on a single lidar."

to be

"The assumption is robust so long as uh ≫ w tan φ. Significant vertical velocities do occur, given the complex terrain and wake, and can lead to error in the estimate due to the inclusion of non-negligible projection of the vertical velocity. When relying on a single lidar, limiting the elevation angles of the beams can help to constrain the degree of contamination from vertical velocities. Here, the beams scanning the upwind ridge in the area of the wake typically have elevations angles φ < 20◦ (tan 20◦ ≈ 0.36)."

16. Pg. 10, line 195: ". . . any sweep with time stamps inside the window are included." What does this phrase mean? Please make this statement clear.

We have clarified the original statement

"Only full RHI lidar sweeps are used, and any sweep with time stamps inside the window are included"

to say more explicitly that

"Only full RHI lidar sweeps are used; for a given 5-minute window, if any beams have a timestamp within the window, the whole sweep is included in the average."

17. Pg. 10, line 211: "Maximum drag on . . . the blade tip, . . . " From Fig. 4(a), it seems maximum velocity deficit occurs around the middle of the blade. This does not agree with your statement.

Thank you for catching our typo; we have fixed this to read "Maximum drag on the wind field typically occurs midway along the blade…".

18. Pg 13, line 271 through 281: Are you trying to discuss the results in Fig. 5 over here?

No; these two paragraphs give a general description of the flow behavior during the simulation (we call out Fig. 3 to help illustrate) and establish the smaller error in the unwaked transect which is not shown. Fig. 5 is explicitly and correctly introduced in the next paragraph and further discussed in more detail where the panels are individually referenced in the following text (lines 305-311).

In the discussion of the unwaked transect, we have added an explicit note to indicate that an associated figure is not included.

"In the unwaked transect, the single DTU lidar captures the background flow well, placing the mountain wave and recirculation zones accurately and reproducing all but the most extreme velocity magnitudes (not shown)."

19. Pg. 13, line 292: "By design, . . . " You cannot make such a statement about elevation angle, until you compare results from two or more elevation angles?

As you suggested in comment #15, the underlying LES flow field provides a 'truth' from which we explicitly calculate the error due to the projection onto the elevated beams – this explicit error is what is shown in Fig. 5(b) and the basis of the statement about the magnitude of the error due to the projection. The statement about the growth of the error with height refers to the larger values in the error profiles higher up where the flow is probed with beams at greater elevation angles. This is in alignment with the form of the error due to the projection of the vertical velocity onto the angled beams, given the last term of eq 7,

$$u_{h,lidar} = v_r \cos \varphi = u_h + \mathbf{w \tan \varphi} \quad (7)$$

We have reworded the original sentence:

"By design, the elevation angle of the beams is kept small, which proves effective at limiting the error contribution from the projection of the vertical velocity to generally less than 0.2 m/s near the wake, predictably growing with height and the elevation of the beam (Fig. 5(b))."

to try to clarify that we are showing and discussing error computed from the model and

"The error contribution from the projection of the vertical velocity is generally less than 0.2 m/s near the wake (Fig. 5(b)), where the magnitude of the error grows higher up in the profiles where the elevation angles of the beams are greater."

20. Figure 6 (a): The wake width drops significantly downstream from 3D. This seems unnatural to me. Also, two lobes disappear too early (≤1.5D) compared to 4 to 5D in many wind tunnel studies. Is it solely due to "insufficient resolution of retrieval points" as stated by the author OR complex terrain is also a contributing factor? This should be clarified in the manuscript.

We understand that these measurements from a real field experiment in complex terrain may differ from wake behavior in the controlled environment of a wind tunnel, but other field measurements also show the two lobes of the wake disappearing faster in the real world than in wind tunnel measurements. For example, Figure 17 of Aitken et al. 2014 (https://journals.ametsoc.org/view/journals/atot/31/4/jtech-d-13-00104_1.xml) shows that even at 1 rotor diameter downwind, one third of the wakes detected are single-lobe wakes. The field measurements of Zahn et al. 2020 (https://onlinelibrary.wiley.com/doi/epdf/10.1002/we.2430), Figure 12, show few two-lobe wakes at distances after 1.75D, and the averaged profiles in Figure 15A-D of Zhan et al. also show the two-lobe structure is gone after 2D. We respectfully suggest that expecting field measurements of wakes over complex terrain in a turbulent atmosphere to conform to wake measurements in the controlled environment of a wind tunnel is not realistic and we have reported metrics reflecting the behavior present in our study. While complex terrain will impact the wake, these effects are captured in the 'true' LES wake on which the lidar

retrievals are made and statements about the insufficient resolution of retrieval points only refer to the retrieval of the LES wake.

21. Pg. 16, line 319: What is "peak velocity deficit magnitude" in Fig. 7? Please define it...if necessary using an equation.

The parameters derived from the fitting of the wake, including the maximum velocity deficit magnitude" are discussed in Sect. 4.2 with the wake fitting algorithm. In this section it is defined to be the largest value in the fitted velocity deficit profile, i.e. the maximum amount the flow slowed due to the wake ; its specific definition for the fitted Gaussian models are given in the text immediately after eq (9) and (10).

To make the connection with the original introduction of the parameter more clear, we have updated the wording of "peak velocity deficit magnitude" to "maximum velocity deficit magnitude" for consistency here and in two other instances where 'peak' was used.

22. Fig. 7: What is the definition of the "magnitude of the wake deficit". Please add its mathematical definition (or description) in the text.

Please see Sect. 4.2 (see previous response). We have updated "peak LES wake deficit" in the caption to "maximum LES wake deficit magnitude" to make the terminology consistent.

23. How can you relate simulation results (Fig. 7, 8, 9) against the observations in Fig. 10? This is not clear from the discussion in section 5.1 and 5.2.

We leverage the fact that there are two independent systems (DLR1 and DLR2) measuring the wake so that we can compare the wake retrievals of the two both with real and virtual measurements as described in lines 364-367. We have added to line 366 a reference to where this intercomparison is first shown for the virtual measurements.  "A similar inter-comparison can be done using the two virtual DLR instruments (as shown in panel (b) of Fig. 7-9)".

A discussion of how the observational behavior compares to what is expected from the simulation results is given in lines 391-402.

24. Pg. 21, line 439 through 441: I do not think you have verified these statements regarding elevation angle. For example, you could have done scan at higher elevation angle and compared the results against 20∘ elevation limit you have set.

As we have hopefully clarified in our responses to comments #15 and #19, the 20∘ is not a set limit but was only used in describing a rough range of beam angles scanning in the area of the wake; statements about the error associated with the beam elevation angles are derived from the results explicitly treating the error calculated with respect to the true LES velocities (see lines 309-311 and Fig 5(b)).

 "For the Perdigão case, the limitation of the beam elevation angles around the wake prove largely effective at minimizing error due to contamination by projected vertical velocities."

As shown in Fig. 5(b), even though the vertical velocities are nonnegligible (reaching nearly 4 m/s in some places), the errors due to the projection of the vertical velocity are typically smaller than <0.2 m/s around the wake. The elevation angle tan φ weighting on the vertical velocity in the horizontal velocity calculation is what attenuates the values; given that we explicitly show these errors to be smaller than other error contributions in the system, we feel comfortable in saying that the 'limitation of the beam elevation angles around the wake prove largely effective at minimizing error' for this case.

25. Pg. 22, line 445 through 448: You can move the literature review to Introduction. I do not see the point in putting this paragraph over here.

We have omitted this paragraph and kept just the final statement statement that "This virtual lidar tool can help enable quantification of possible errors due to scanning geometries and scanning

strategies, to enable optimal field experiment planning and instrument deployment." as part of the preceding paragraph.

**Minor comments and corrections:**

Pg2, line 48: Section 3 is missing.

Thank you for catching this omission. We've corrected the original sentence,

"Section 2 describes the Perdigão site, selected case studies and data processing, as well as the setup of the LES and virtual lidar models."

to call out section 3,

"Section 2 describes the Perdigão site, selected case studies and data processing, with the setup of the LES and virtual lidar models in Sect. 3."

Table 1: Date format ins the 1st and the 2nd rows should be same. Thank you for catching this inconsistency in formatting; corrected.

Also, add standard deviation of wind direction for all three cases.

Alongside mean values reported in Table 1, we have added the standard deviations of the wind speed and direction using 1-minute averaged data for each of the cases.

Pg. 4, line 67: "three towers . . ." I can only see two towers in Fig. 1. Where is the third tower.

Thanks for catching the omission; the figure has been fixed to include tower 13.

Pg. 5, line 92: Do you mean CNR > −24dB? Yes; thank you for catching. Corrected.

Pg. 15, line 305 through 314: Should be a single paragraph. Done.

Pg. 20, line 403: ". . . at near. . ." Remove 'at'. Corrected.

---

## Author Response (AR2)

Thank you to the reviewer for their careful attention to the references. We have corrected the duplicates in the resubmitted manuscript and *.tex files. The submitted files also reflect the inclusion of an updated affiliation for Julie K. Lundquist.